# RESEARCH

# A comprehensive and quantitative exploration of thousands of viral genomes

**Abstract** The complete assembly of viral genomes from metagenomic datasets (short genomic sequences gathered from environmental samples) has proven to be challenging, so there are significant blind spots when we view viral genomes through the lens of metagenomics. One approach to overcoming this problem is to leverage the thousands of complete viral genomes that are publicly available. Here we describe our efforts to assemble a comprehensive resource that provides a quantitative snapshot of viral genomic trends – such as gene density, noncoding percentage, and abundances of functional gene categories – across thousands of viral genomes. We have also developed a coarse-grained method for visualizing viral genome organization for hundreds of genomes at once, and have explored the extent of the overlap between bacterial and bacteriophage gene pools. Existing viral classification systems were developed prior to the sequencing era, so we present our analysis in a way that allows us to assess the utility of the different classification systems for capturing genomic trends.
DOI: https://doi.org/10.7554/eLife.31955.001

**GITA MAHMOUDABADI AND ROB PHILLIPS\***

## Introduction

There are an estimated $10^{31}$ virus-like particles inhabiting our planet, outnumbering all cellular life forms (*Suttle, 2005*; *Wigington et al., 2016*). Despite their presence in astonishing numbers and their impact on the population dynamics and evolutionary trajectories of their hosts, our quantitative knowledge of trends in the genomic properties of viruses remains largely limited with many of the key quantities used to characterize these genomes either scattered across the literature or unavailable altogether. This is in contrast to the growing ability exhibited in resources such as the BioNumbers database (*Milo et al., 2010*) to assemble in one curated collection the key numbers that characterize cellular life forms. Our goal has been to complement these databases of key numbers of cell biology (*Milo et al., 2010*; *Phillips et al., 2012*; *Milo and Phillips, 2015*; *Phillips and Milo, 2009*) with corresponding data from viruses. With the advent of high-throughput sequencing technologies, recent studies have enabled genomic and metagenomic surveys of numerous natural habitats, untethering us from the organisms we know and love and giving us access to a sea of genomic data from novel organisms (*Paez-Espino et al., 2016*). Such advances allow us to appreciate the genomic diversity that is a hallmark of viral genomes (*Paez-Espino et al., 2016*; *Edwards and Rohwer, 2005*; *Rohwer and Thurber, 2009*; *Simmonds et al., 2017*; *Simmonds, 2015*; *Mokili et al., 2012*) and now make it possible to assemble some of the key numbers of virology.

In contrast to cellular genomes, which are universally coded in the language of double-stranded DNA (dsDNA), genomes of viruses are remarkably versatile. Viral genomes can be found as single or double-stranded versions of DNA and RNA, packaged in segments or as one piece, and present in both linear and circular forms. Additionally, based on their rapid infectious cycles, large burst sizes, and often highly error-prone replication, viruses collectively survey a large genomic sequence space, and comprise a great portion of the total genomic diversity hosted by our planet (*Kristensen et al.,*

**\*For correspondence:** phillips@ pboc.caltech.edu

**Competing interests:** The authors declare that no competing interests exist.

2010; Hendrix, 2003). Recently, through a large study of metagenomic sequences, the known viral sequence space was increased by an order of magnitude (Paez-Espino et al., 2016), and much more of the viral "dark matter" likely remains unexplored (Youle et al., 2012).

In analyzing an increasing spectrum of sequence data, we are faced with a considerable challenge that is unique to viruses, namely, how to find those features within viral genomes that might reveal hidden aspects of their evolutionary history. To put this challenge in perspective, when analyzing non-viral data, universal markers from the ribosomal RNA such as 16S sequences are used to classify newly discovered organisms and to locate them on the evolutionary tree of life (Hug et al., 2016). Virus genomes on the other hand are highly divergent and possess no such universally shared sequences (Kristensen et al., 2011).

In the absence of universal genomic markers, viruses have historically been classified based on a variety of attributes, perhaps most notably morphological characteristics, proposed in 1962 by the International Committee on Taxonomy of Viruses or ICTV (King et al., 2011), or based on the different ways by which they produce mRNA, proposed by David Baltimore in 1971 (Baltimore, 1971; Figure 1). The ICTV classifies viruses into seven orders: Herpesvirales, large eukaryotic double-stranded DNA viruses; Caudovirales, tailed double-stranded DNA viruses typically infecting bacteria; Ligamenvirales, linear double-stranded viruses infecting archaea; Mononegavirales, nonsegmented negative (or antisense) strand single-stranded RNA viruses of plants and animals; Nidovirales, positive (or sense) strand single-stranded RNA viruses of vertebrates; Picornavirales, small positive strand single-stranded RNA viruses infecting plants, insects, and animals; and finally, the Tymovirales, monopartite positive single-stranded RNA viruses of plants. In addition to these orders, there are ICTV families, some of which have not been assigned to an ICTV order. Only those ICTV viral families with more than a few members present in our dataset are explored.

The Baltimore classification groups viruses into seven categories (Figure 1): double-stranded DNA viruses (Group I); single-stranded DNA viruses (Group II); double-stranded RNA viruses (Group III); positive single-stranded RNA viruses (Group IV); negative single-stranded RNA viruses (Group V); positive single-stranded RNA viruses with DNA intermediates (Group VI), commonly known as retroviruses; and, the double-stranded DNA retroviruses (Group VII).

Given the prevalence of these viral classification systems in the categorization of viruses today, it is worth remembering that their inception predates the sequencing of the first genome in 1976. With the fastest and cheapest rates of sequencing available to date, we live at an opportune moment to explore viral genomic properties and evaluate these existing classification systems in light of the growing body of sequence information.

In addition to the ICTV and the Baltimore classifications we used a simple classification system based on the host domain information, and divided viruses into bacterial, archaeal and eukaryotic viruses (Figure 1). The underpinning motivation behind this kind of classification is the Coevolution Hypothesis (Mahy and Van Regenmortel, 2010; Forterre, 2010). Viruses are obligate organisms unable to survive without their host, and as a corollary it is hypothesized that they have coevolved with their hosts as the hosts diverged over billions of years to form the three domains of life (Mahy and Van Regenmortel, 2010; Forterre, 2010). A possible piece of supporting evidence for this hypothesis is that there are to date no reported infections of hosts from one domain by viruses of another observed. We also explored a minimal classification system that divides the virus world into two groups based on their nucleotide type (RNA and DNA), here termed "Nucleotide Type" classification (Figure 1). This classification is introduced as a simplified version of the Baltimore classification system. In practice, we have assigned Baltimore groups 1, 2 and 7 to the DNA viral category, and the remaining Baltimore groups to the RNA viral category.

Although many viruses are uncharacterized, at the time of the analysis of the data presented here, there were 4,378 completed genomes available from the NCBI viral genomes resource (Brister et al., 2015) (data acquired in August, 2015). However, large-scale analyses of genomic properties for these viruses are generally unavailable. This stands in stark contrast to the in-depth analyses performed on partially assembled viral genomes or viral contigs derived from metagenomic studies (Paez-Espino et al., 2016; Roux et al., 2016). Although these studies have uncovered many important aspects of viral ecology with relatively little bias in sampling, they are limited by the fact that metagenomic studies typically do not result in the full assembly of genomes. An interesting example that illustrates

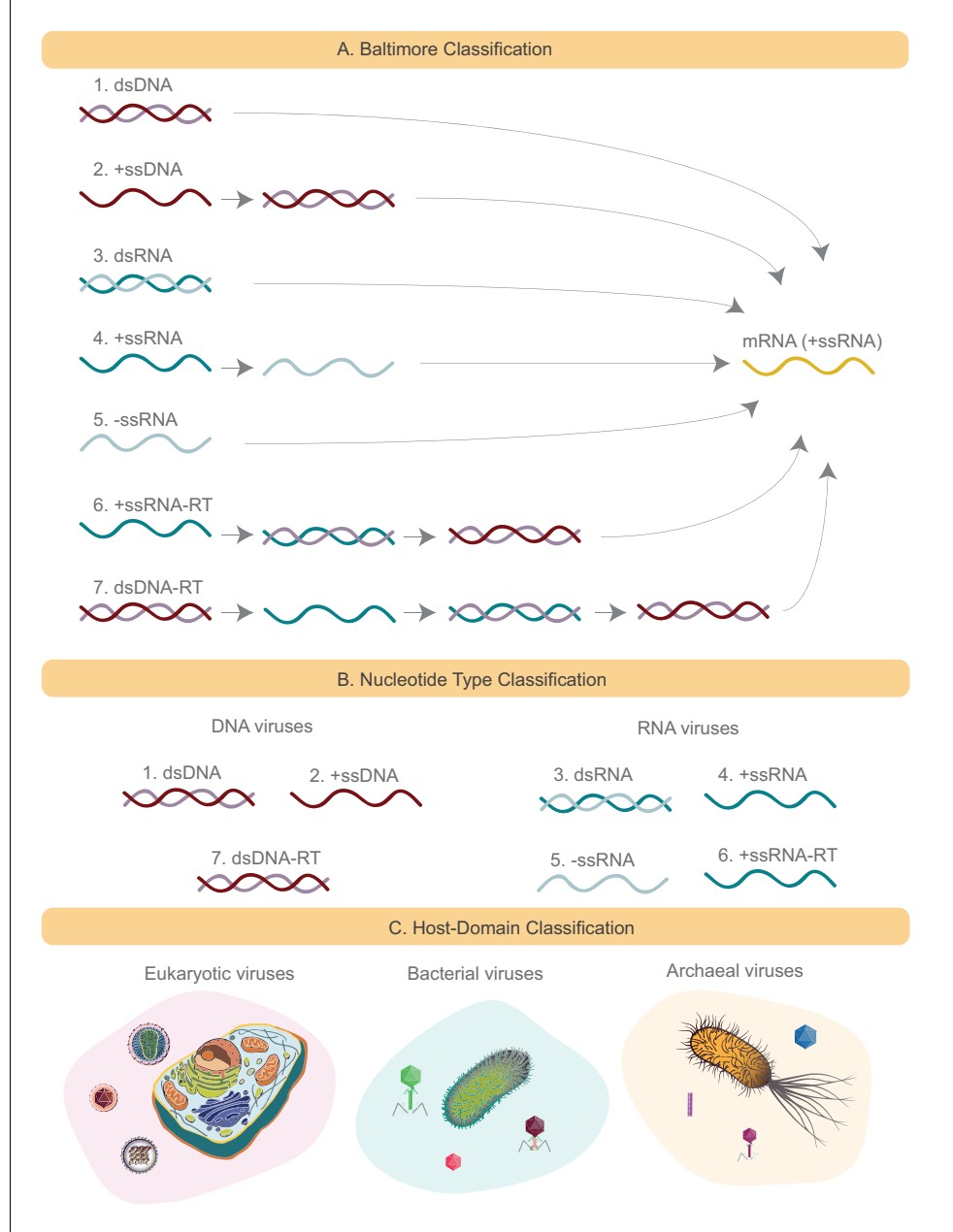

**Figure 1.** Schematics of several viral classification systems explored in this study. (**A**) The Baltimore classification divides all viruses into seven groups based on how the viral mRNA is produced. DNA strands are denoted in red (+ssDNA in darker shade of red than -ssDNA). Similarly RNA strands are denoted in green (+ssRNA in darker shade of green than -ssRNA). In the case of Baltimore groups 1,2,6, and 7, the genome either is or is converted to dsDNA, which is then converted to mRNA through the action of DNA-dependent RNA polymerase. In the case of Baltimore groups 3, 4 and 5, the genome is or is converted to +ssRNA, which is mRNA, through the action of RNA-dependent RNA polymerase. (**B**) Nucleotide type classification divides viruses based on their genomic material into DNA and RNA viruses. Baltimore viral groups 1, 2, and 7 are all considered DNA viruses, and the remaining viral groups are considered RNA viruses. (**C**) Host Domain classification groups viruses based on the host domain that they infect. Three groups are formed: eukaryotic, bacterial and archaeal viruses.
DOI: https://doi.org/10.7554/eLife.31955.002

the difficulty of complete genome assembly from metagenomic studies is the crAssphage genome, which despite taking prominent fractions of reads across various metagenomic datasets, had gone undetected and remained unassembled (*Dutilh et al., 2014*). However,

recent methods to counter these limitations provide a promising future for the use of metagenomic datasets in capturing complete genomes from complex environments (*Marbouty et al., 2017*; *Nielsen et al., 2014*).

Without complete viral genomes, it would be difficult to develop systematic understanding of key aspects of viral genomic architecture. To address this problem at least in part, we set out to provide a large-scale analysis of various genomic metrics measured from existing complete viral genomes. To perform a comprehensive analysis, we first explored the diversity of known viruses and their hosts within the NCBI database (see Materials and methods). We then created distributions on a number of metrics, namely genome length, gene length, gene density, percentage of noncoding DNA (or RNA), functional gene category abundances, and gene order. We have provided brief introductions to these metrics in the following subsections.

### Viral genome length, gene length and gene density

Genomes are replete with information about an organism's past and present. A central and revealing piece of information is the genome length. As more and more complete genomes have become available, we have learned that genome lengths of cellular organisms vary quite extensively, specifically by six orders of magnitude (*Phillips et al., 2012*; *Alberts et al., 2002*). Because these studies focused on cellular organisms, and because genome length information is generally inaccessible through metagenomic studies, large-scale analyses that systematically capture viral genome length distributions in light of different classification systems and in relation to other genomic parameters are lacking. One such genomic parameter is the number of genes that are encoded per genome, also referred to as gene density (*Keller and Feuillet, 2000*; *Hou et al., 2012*). Another set of missing distributions involves gene lengths, and here too, it is important to see how they vary across different viral classification categories.

### The noncoding percentages of viral genomes

One of the most surprising discoveries of the past several decades was the rich and enormous diversity of noncoding DNA in the human genome (*Elgar and Vavouri, 2008*). Though originally thought of as "junk DNA", the noncoding regions of our genomes were later shown to be of great functional importance. Noncoding DNA is an umbrella term for very different elements, for example functional RNAs such as micro RNAs (miRNA), regulatory elements such as promoters and enhancers, as well as transposons and pseudogenes.

Moreover, genomes vary widely in their noncoding percentages. While multicellular eukaryotic genomes such as plants and vertebrates have 50% or more of their genomes filled with noncoding regions, single-cell eukaryotic genomes have 25-50% of their genomes present as noncoding regions and prokaryotic genomes have even lower percentages of noncoding DNA, generally 15 to 20% (*Mattick and Makunin, 2006*; *Morris, 2012*; *Mattick, 2004*). Hence, the noncoding percentage of the genome is thought to correlate with the phenotypic complexity of the organism, and consequently, much of the investigation into noncoding fractions of genomes has been focused on higher eukaryotes. However, the discovery of the bacterial immunity against phages and other sources of foreign DNA, otherwise known as CRISPR/Cas system (Clustered Regularly Interspaced Short Palindromic Repeats), as well as the discovery of a new class of antibiotics targeting bacterial noncoding DNA (*Howe et al., 2015*) demonstrate the level of biotechnological impact and scientific insight that the study of noncoding elements in bacteria can provide. Even less is known about the noncoding fraction of viral genomes.

The literature on viral noncoding DNA or RNA is relatively sparse but highly intriguing. The first viral noncoding RNAs were discovered in adenoviruses, dsDNA viruses that infect humans, and were ~160 base pairs long (*Reich et al., 1966*; *Tycowski et al., 2015*; *Steitz et al., 2011*). These sequences were shown responsible for viral evasion of host immunity by inhibition of protein kinase R- a cellular protein responsible for the inactivation of viral protein synthesis (*Mathews and Shenk, 1991*). In ovine herpesvirus, miRNAs have been shown to maintain viral latency (*Riaz et al., 2014*). These are just several examples in which viral noncoding elements have been shown to enable viral escape from host immunity, as well as regulate viral life-cycle and viral persistence (*Tycowski et al., 2015*). Despite many interesting studies exploring the topic of cellular noncoding DNA (*Mattick and Makunin, 2006*; *Morris, 2012*; *Mattick, 2004*), there are no studies, to our knowledge, that reveal the statistics of noncoding percentage of viral genomes.

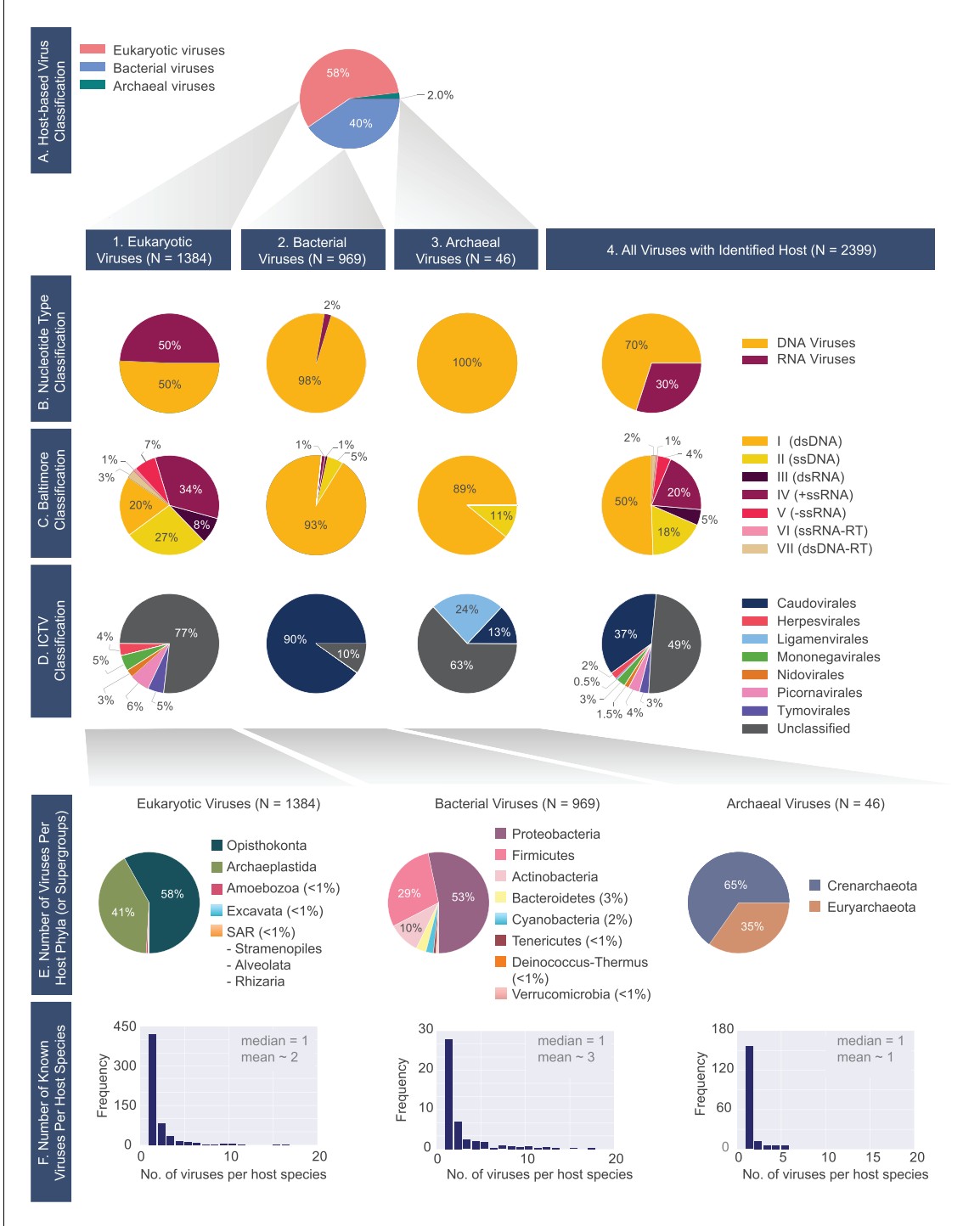

**Figure 2.** A census of all viruses with complete genomes reported to NCBI that were matched to a host (N= 2399). (A) Percentage of viruses infecting hosts from the three domains of life. 1) Eukaryotic, 2) bacterial and 3) archaeal viromes are further classified according to the (B) Nucleotide Type, (C) Baltimore, and D) ICTV classification systems. (E) Distributions of host phyla (or supergroups) infected by the (1) eukaryotic, (2) bacterial, and (3) archaeal viruses is shown. As in the case of panel F, the host taxonomic identification is derived from the NCBI Taxonomy database (see Materials and methods). (F) Histograms of the number of known viruses infecting host species. Median and mean number of viruses infecting a host species is provided in each plot. The full-range of x-values for the bacterial and eukaryotic histograms extends beyond n=20 (see virusHostHistograms.ipynb in our GitHub repository [*Mahmoudabadi, 2018*]). Further exploration of the largest fraction of the eukaryotic virome (i.e. animal viruses) is shown in *Figure 2—figure supplement 1*.

DOI: https://doi.org/10.7554/eLife.31955.003

*Figure 2 continued on next page*

*Figure 2 continued*

The following figure supplement is available for figure 2:

**Figure supplement 1.** Further exploration of the largest fraction of the eukaryotic virome: viruses of Opisthokonta supergroup (animals).
DOI: https://doi.org/10.7554/eLife.31955.004

### Viral functional gene categories

There are detailed studies on the counts of cellular genes belonging to each broad functional category (*Molina and van Nimwegen, 2009*; *Grilli et al., 2012*). These studies have helped us better understand the scaling of functional categories across different clades of organisms. In fact there was an intriguing conclusion that for prokaryotic genomes, there exists a universal organization which governs the relative number of genes in each category (*Molina and van Nimwegen, 2009*). Such depictions of viral genomes, however, are largely lacking. Thus, we set out to better understand how viral genes are distributed across different functional categories and how these distributions might differ across various viral groups.

### Viral genome organization

Viral genome organization is a topic that has great depth but limited breadth. There exist highly detailed genome-wide diagrams that illustrate the location, direction, and predicted function of viral genes, which are then compared to similar illustrations from a small number of viral genomes (*Labonté et al., 2015*; *Casjens et al., 2005*; *Marinelli et al., 2012*; *Brüssow and Hendrix, 2002*). While this highly detailed approach is indispensible for studying individual viruses, a simplified illustration of genome organization is a requirement of any high-throughput visualization and comparison of genomes. The latter approach could help us uncover general rules governing genomic organization, in the same way that synteny, or conserved gene order, has been used to compare animal genomes (*Telford and Copley, 2011*; *Jaillon et al., 2004*) and genomes of RNA viruses infecting invertebrates (*Shi et al., 2016*).

## Results

### Exploring the NCBI viral database

We used the largest available dataset of completed viral genomes available from the National Center for Biotechnology Information (NCBI) viral genomes resource (*Brister et al., 2015*), containing a total of 4,378 complete viral genomes at the time of data acquisition (August,

2015). After implementing several manual and programmed steps towards curating the data, a total of 2,399 viruses (excluding satellite viruses) could be associated with a host using NCBI's documentation (see Materials and methods). These viruses were included for further analysis, and unless noted otherwise, will constitute our dataset in this study. By examining these viruses through different classifications (*Figure 2*), it is clear that they are largely DNA viruses (*Figure 2B4*), and more specifically, they are primarily double-stranded DNA (dsDNA) viruses (*Figure 2C4*). This is in contrast to the RNA viruses in this database, which are mostly single-stranded (*Figure 2B4* and *Figure 2C4*).

We further observed that eukaryotes host nearly an equal number of DNA and RNA viruses (*Figure 2B1*). In contrast to prokaryotes, which are predominantly host to viruses with double-stranded genomes, eukaryotes are host to a higher number of viruses with single-stranded genomes. Why are double-stranded DNA viruses, despite their high prevalence in the bacterial and archaeal world, only the third largest group of viruses infecting eukaryotes in this database? One explanation proposed is the physical separation of transcriptional processes from the cytoplasm by way of the eukaryotic nucleus (*Koonin et al., 2015*). This physical separation is thought to impose an additional barrier for DNA viruses in gaining access to the host's transcriptional environment.

More than half of viruses with complete genomes have not been assigned to any viral orders under the ICTV classification (*Figure 2D4*). About one third of all known viruses are assigned to the *Caudovirales* order, while the other orders are in the minority. The vast majority of the bacterial viruses are categorized as part of the *Cauodvirales* order (*Figure 2D2*), but the majority of archaeal and eukaryotic viruses remain unassigned to any order.

Before any further exploration of this dataset, we aimed to assess its diversity and possible sources of bias (*Figure 2E–F*). It was immediately clear, for example, that archaeal viruses were heavily under-sampled. In contrast, bacterial viruses infect hosts from a diverse array of

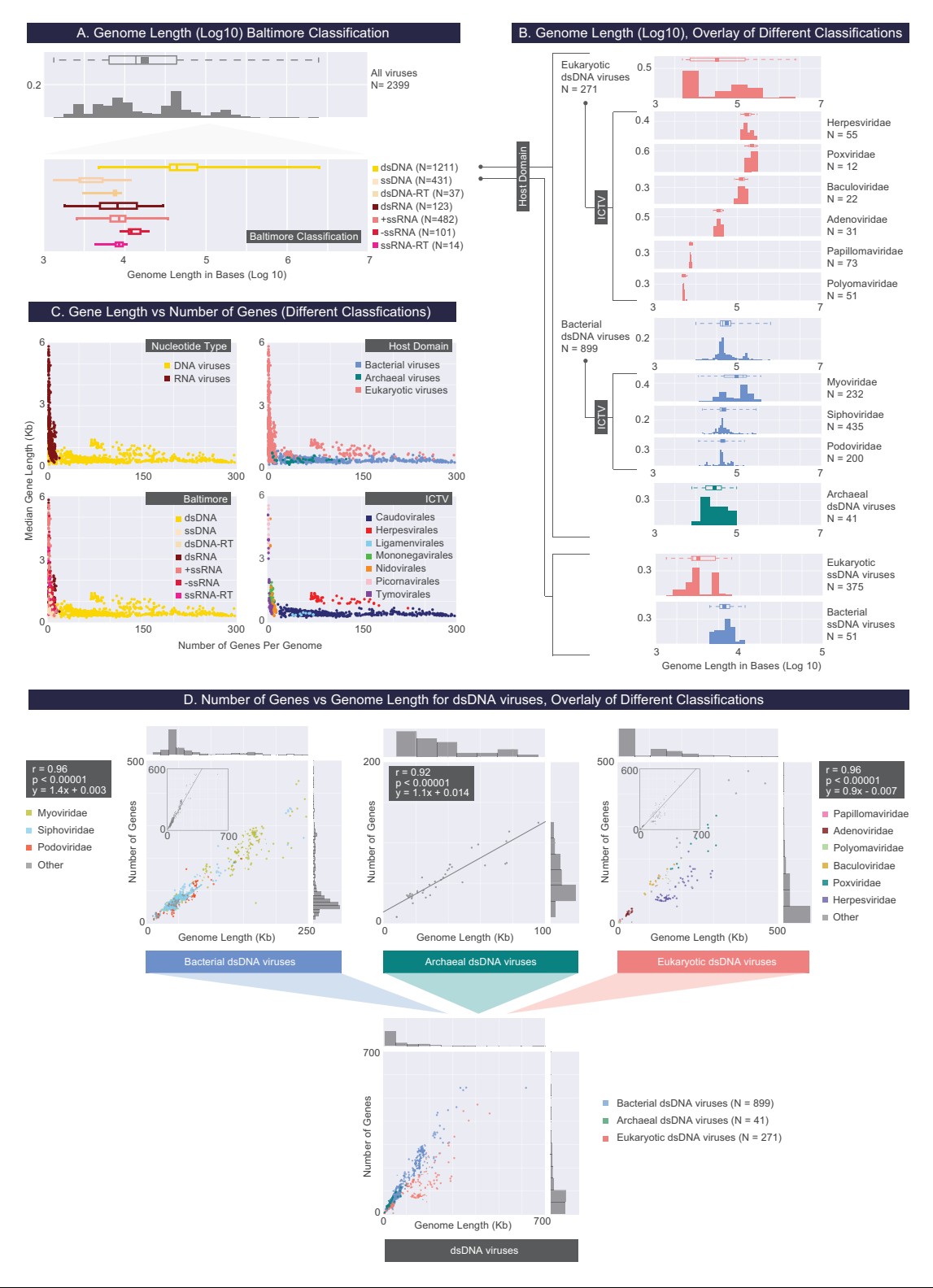

**Figure 3.** Describing viral genomes through distributions of genome length, gene length and gene density. (A) Box plots of genome lengths (Log10) across all viruses included in our dataset (top), further partitioned based on the Baltimore classification categories (bottom). The number of viruses included in each group is denoted by N. (B) A closer examination of dsDNA and ssDNA viral genome lengths through the overlay of Host Domain and ICTV classification systems. Distributions of genome lengths associated with eukaryotic, bacterial and archaeal viruses are shown in salmon, blue, and teal, respectively. ICTV viral families with only a few members are omitted. Distributions of genome lengths across different classification systems along

*Figure 3 continued on next page*

*Figure 3 continued*

with various statistics are shown in *Figure 3—figure supplement 1*. and *Figure 3—source data 1*. Note that the bimodal distribution of eukaryotic ssDNA viruses, which also appears in the next figure, arises from the Begomoviruses, which are plant viruses with circularized monopartite and bipartite genomes (*Melgarejo et al., 2013*). (C) Median gene length is plotted against the number of genes for each genome for all genomes in our dataset, color-coded according to different classification systems. (D) Number of genes per genome length (gene density) for dsDNA viruses based on the overlay of Host Domain (bottom) and ICTV family classification categories (top) (Pearson correlations and their statistical significance, two-tailed t-test P values, are denoted).

DOI: https://doi.org/10.7554/eLife.31955.005

The following source data and figure supplement are available for figure 3:

**Source data 1.** Genome length statistics for viral groups across different classification systems (rounded to the nearest kilobase).
DOI: https://doi.org/10.7554/eLife.31955.007
**Figure supplement 1.** Histograms of genome length (Log10) across all complete viral genomes associated with a host.
DOI: https://doi.org/10.7554/eLife.31955.006

bacterial phyla (*Figure 2E2*). However, even for bacterial viruses, there are host phyla whose viruses are entirely missing from the database, for example *Synergistes* and *Acidobacteria*, whose members are typically unculturable soil bacteria. Given that the isolation and characterization of archaeal and bacterial viruses has traditionally been dependent on the culturing of their hosts, the majority of viruses with unculturable hosts remain unexplored. Moreover, the eukaryotic viruses in the database infect hosts primarily from the *Viridiplantae* or the *Opisthokonta* supergroups (*Figure 2E1*). Among *Viridiplantae*, the majority of hosts belong to the *Streptophytina* group (land plants), and within the *Opisthokonta* supergroup, the majority of viruses are metazoan. We further examine the distribution of viruses from the *Opisthokonta* supergroup in *Figure 2—figure supplement 1*.

We continued to explore host diversity at a finer resolution and mapped out the number of viruses that infect each host species (*Figure 2F*). As expected, organisms such as *Staphylococcus aureus*, *Escherichia coli*, and *Solanum lycopersicum*, which are host species with either medical, research or agricultural relevance, have many known viruses and are outliers in the skewed distributions shown in *Figure 2F*. However, the median number of viruses known to infect a eukaryotic or a prokaryotic host species is approximately 1 (*Figure 2F*). This signifies that even for host species that are already represented in our collection, the number of known viruses is likely an underestimate considering the larger numbers of viruses known to infect the more heavily studied host species.

### Viral genome lengths, gene lengths, gene densities

Genome lengths for all fully sequenced viral genomes varied widely by three orders of magnitude (*Figure 3A*, *Table 1*). According to the Host Domain classification, prokaryotic viruses tend to have longer genomes than eukaryotic viruses (*Figure 3—source data 1*, *Figure 3—figure supplement 1*). However, this difference can be better explained by the Nucleotide Type classification, as the median RNA virus genome length is four times shorter than the median DNA virus genome length. Thus, the comparison between prokaryotic and eukaryotic viral genome lengths is confounded by the fact that the prokaryotic virome, as represented by this database, is primarily composed of DNA viruses, whereas the eukaryotic virome is only half comprised of DNA viruses (*Figure 2C4*).

With respect to viral genome lengths, the Baltimore classification seems to offer the most explanatory power. Knowing whether a viral genome is DNA- or RNA-based already provides a strong indication about viral genome length, especially for RNA viruses where the standard deviation is just a few kilobases (*Figure 3—source data 1*). However, by distinguishing between ssDNA, dsDNA and dsDNA-RT viruses, the Baltimore classification offers a more complete view of genome length distributions compared to the binary Nucleotide Type classification (*Figure 3A*). Across all Baltimore groups, dsDNA viruses have genome lengths that have the largest standard deviation, however considering the limited range of genome lengths associated with other Baltimore groups, it is very likely that a larger viral genome will be composed of dsDNA (*Figure 3A*). We provide a more detailed view of genome length distributions by layering different classification systems, first applying the Baltimore classification, followed by the Host Domain and the ICTV family classifications (*Figure 3B*, *Figure 3—source data 1*). Finally, it is worth noting that capsid dimension, surprisingly, does not seem to

correlate with viral genome size, and to different degrees, many viruses are shown to under-utilize the capsid volume (*Brandes and Linial, 2016*).

In viewing the relationship between median gene length and number of genes per viral genome (*Figure 3C*), two different coding strategies become apparent. Namely, compared to DNA viruses, RNA viruses exhibit a large range of gene lengths. This trend is at least in part reflective of the challenges faced by RNA viruses when encountering the requirements of their host's translational machinery (*Firth and Brierley, 2012*). For example, many of the RNA genomes we examined closely contained genes that encode polyproteins, ribosomal slippage (frame-shifting) or codon read-through events, among other non-canonical translational mechanisms.

As in the case of genome lengths, by examining only the ICTV or the Host Domain classifications it would be difficult to draw meaningful conclusions about the observed patterns, and in the case of the Host Domain classification, our conclusions would be confounded by the disproportionate ratio of RNA to DNA viruses that are known to infect each host domain in this database. However, the layering of these classification systems offers new insights, which we will discuss in the following paragraphs.

We follow others (*Keller and Feuillet, 2000*; *Hou et al., 2012*) in defining the gene density of a genome as the number of genes divided by

**Table 1.** Viral genomic statistics based upon different classification systems.

| Classification | | N | Genome length (kb) | Percent noncoding (DNA/RNA) | Median gene length (bases) |
|---|---|---|---|---|---|
| Host Domain | Eukaryotic Viruses | 1384 | 8 | 10 | 1055 |
| | Bacteria Viruses | 969 | 43 | 9 | 408 |
| | Archaea Viruses | 46 | 24 | 10 | 400 |
| Baltimore | Group I (dsDNA) | 1211 | 44 | 9 | 429 |
| | Group II (ssDNA) | 431 | 3 | 14 | 588 |
| | Group III (dsRNA) | 123 | 8 | 8 | 2291 |
| | Group IV (+ssRNA) | 482 | 9 | 5 | 2366 |
| | Group V (-ssRNA) | 101 | 12 | 7 | 1353 |
| | Group VI (ssRNA-RT) | 14 | 8 | 16 | 1799 |
| | Group VII (dsDNA-RT) | 37 | 8 | 11 | 558 |
| Nucleotide Type | DNA Viruses | 1679 | 38 | 10 | 444 |
| | RNA Viruses | 720 | 9 | 6 | 2072 |
| ICTV (orders) | Caudovirales | 879 | 44 | 9 | 408 |
| | Herpesvirales | 55 | 159 | 19 | 1107 |
| | Ligamenvirales | 11 | 37 | 12 | 372 |
| | Mononegavirales | 71 | 12 | 8 | 1266 |
| | Nidovirales | 35 | 27 | 3 | 672 |
| | Picornavirales | 89 | 8 | 11 | 7056 |
| | Tymovirales | 73 | 8 | 4 | 693 |
| Combinations of different classifications | All Eukaryotic dsDNA viruses | 271 | 33 | 11 | 990 |
| | All Bacterial dsDNA viruses | 899 | 44 | 9 | 408 |
| | All Archaeal dsDNA viruses | 41 | 28 | 10 | 396 |
| | All Eukaryotic ssDNA viruses | 375 | 3 | 14 | 732 |
| | All Bacterial ssDNA viruses | 51 | 7 | 14 | 348 |

Only median values are reported in this table. Genome length data is rounded to the nearest kilobase. N corresponds to the number of viruses from which data is obtained.

DOI: https://doi.org/10.7554/eLife.31955.008

the genome length (*Figure 3D*). We further partitioned dsDNA viruses according to the Host Domain and subsequently the ICTV (family) classifications. We observed a strong linear correlation between dsDNA viral genome lengths and the number of genes encoded by these genomes (*Figure 3D*). The mean (and median) gene densities for bacterial, archaeal and eukaryotic dsDNA viral genomes are approximately 1.4, 1.6 and 0.9 genes per kilo basepairs. As illustrated by the slopes of the regression lines, as well as through a nonparametric statistical test performed on eukaryotic and bacterial dsDNA viral gene densities (one-sided Mann-Whitney U test, $P<10^{-5}$), bacterial dsDNA viruses have significantly higher gene densities than their eukaryotic counterparts in this database.

A closer examination of median gene lengths more clearly reveals the significantly longer gene lengths of RNA viruses compared to DNA viruses (one-sided Mann-Whitney U test, $P<10^{-5}$) (*Figure 4*, *Table 1*). By focusing on DNA viruses, and further dividing these viruses based on Baltimore, Host Domain and ICTV (family) classifications, we arrive at an interesting trend. Namely, eukaryotic viruses, whether dsDNA or ssDNA,

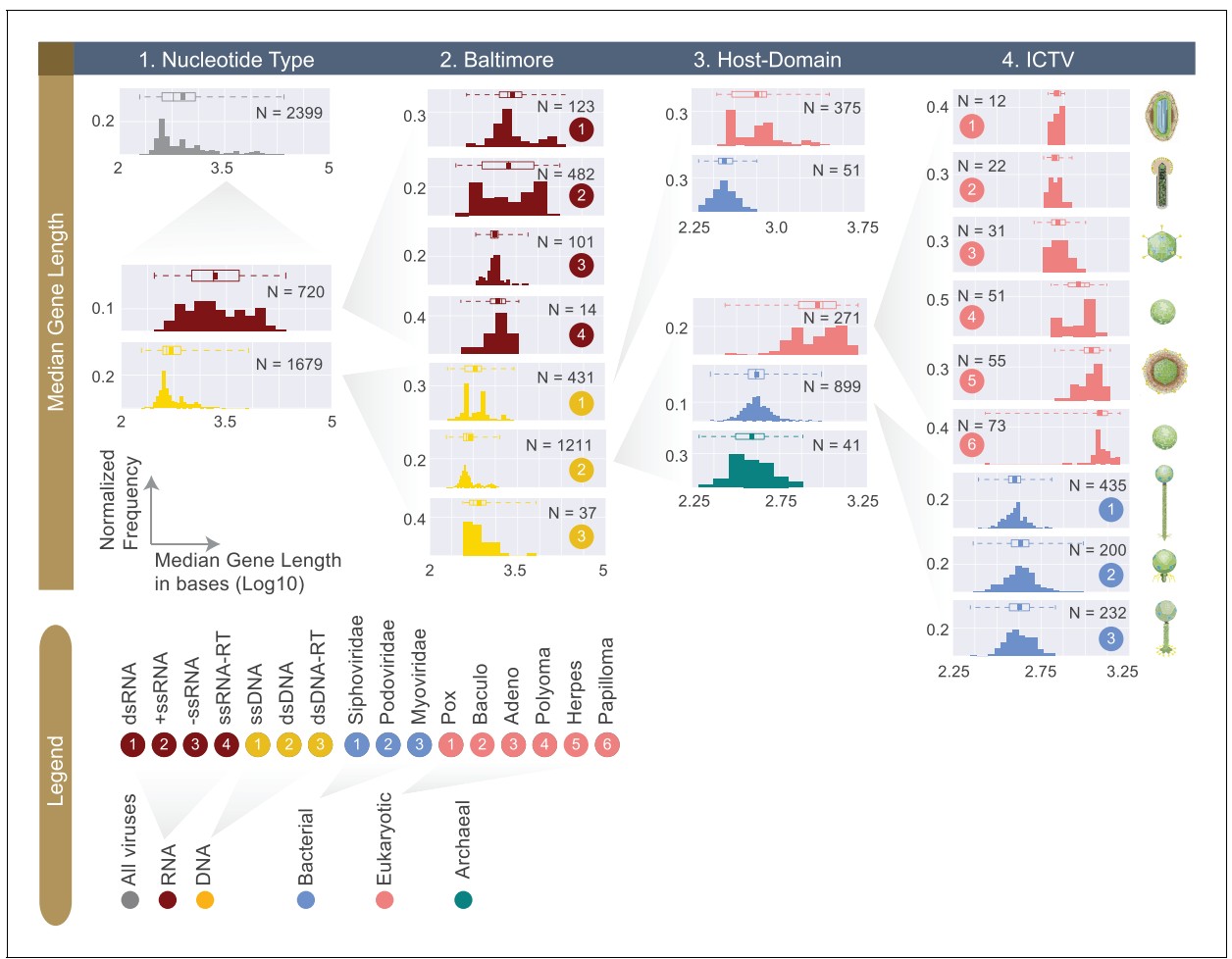

**Figure 4.** Normalized histograms of median gene lengths (log10) across all complete viral genomes associated with a host. Instead of showing absolute viral counts on y-axes, the counts are normalized by the total number of viruses in each viral category (denoted as N inside each plot). The mean of each distribution is denoted as a dot on the boxplot. For all histograms, bin numbers and bin widths are systematically decided by the Freedman-Diaconis rule (*Reich et al., 1966*). Viral schematics on the right of the figure are modified from ViralZone (*Hulo et al., 2011*). Key statistics describing these distributions can be found in *Table 1* and *Figure 4—source data 1*.

DOI: https://doi.org/10.7554/eLife.31955.009

The following source data is available for figure 4:

**Source data 1.** Median gene length statistics for viral groups across different classification systems (rounded to the nearest base).

DOI: https://doi.org/10.7554/eLife.31955.010

have significantly longer gene lengths compared to bacterial viruses from the same Baltimore classification category (*Figure 4*, *Figure 4—source data 1*) (one-sided Mann-Whitney U test, $P<10^{-5}$). This trend follows what we see across cellular genomes, since prokaryotic genes and proteins are shown to be significantly shorter than eukaryotic ones (*Milo and Phillips, 2015*; *Brocchieri and Karlin, 2005*).

## Noncoding percentages of viral genomes

So far we have primarily focused on the coding fractions of viral genomes. Thus, we created distributions of noncoding percentage of viral genomes (see Materials and methods, *Figure 5*, *Table 1*, *Figure 5—source data 1*). In general, DNA viral genomes contain about 10% noncoding regions which is even lower than the noncoding percentage of bacterial genomes

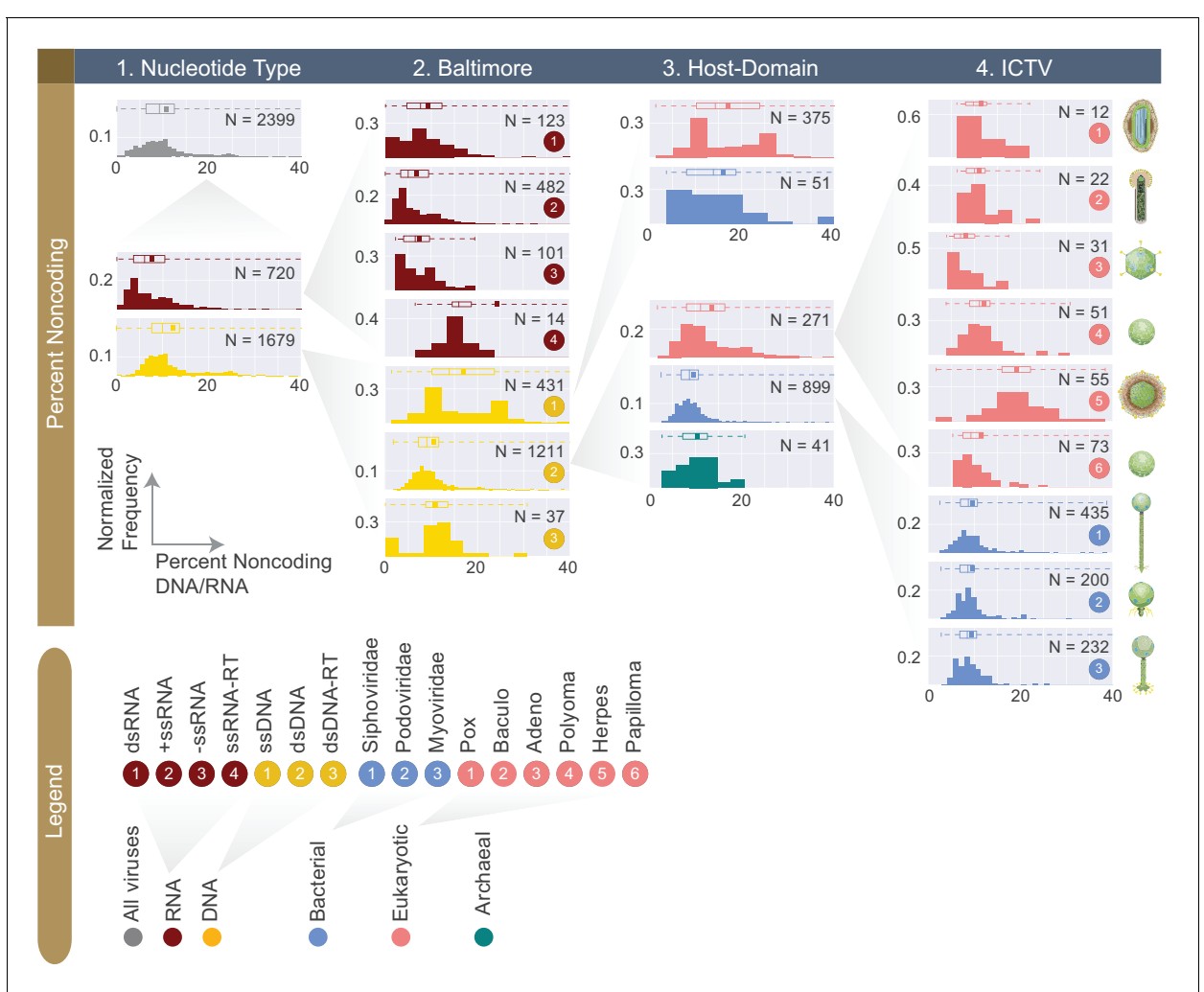

**Figure 5.** Normalized histograms of noncoding DNA/RNA percentage across all complete viral genomes associated with a host. The counts of viruses are normalized by the total number of viruses in each viral category (denoted as N inside each plot). The mean of each distribution is denoted as a dot on the boxplot. For all histograms, bin numbers and bin widths are systematically decided by the Freedman-Diaconis rule (*Reich et al., 1966*). Viral schematics are modified from ViralZone (*Hulo et al., 2011*). Key statistics describing these distributions can be found in *Table 1* and *Figure 5—source data 1*.

DOI: https://doi.org/10.7554/eLife.31955.011

The following source data is available for figure 5:

**Source data 1.** Percent noncoding DNA (or RNA) for viral groups across different classification systems (rounded to the nearest percentage).
DOI: https://doi.org/10.7554/eLife.31955.012

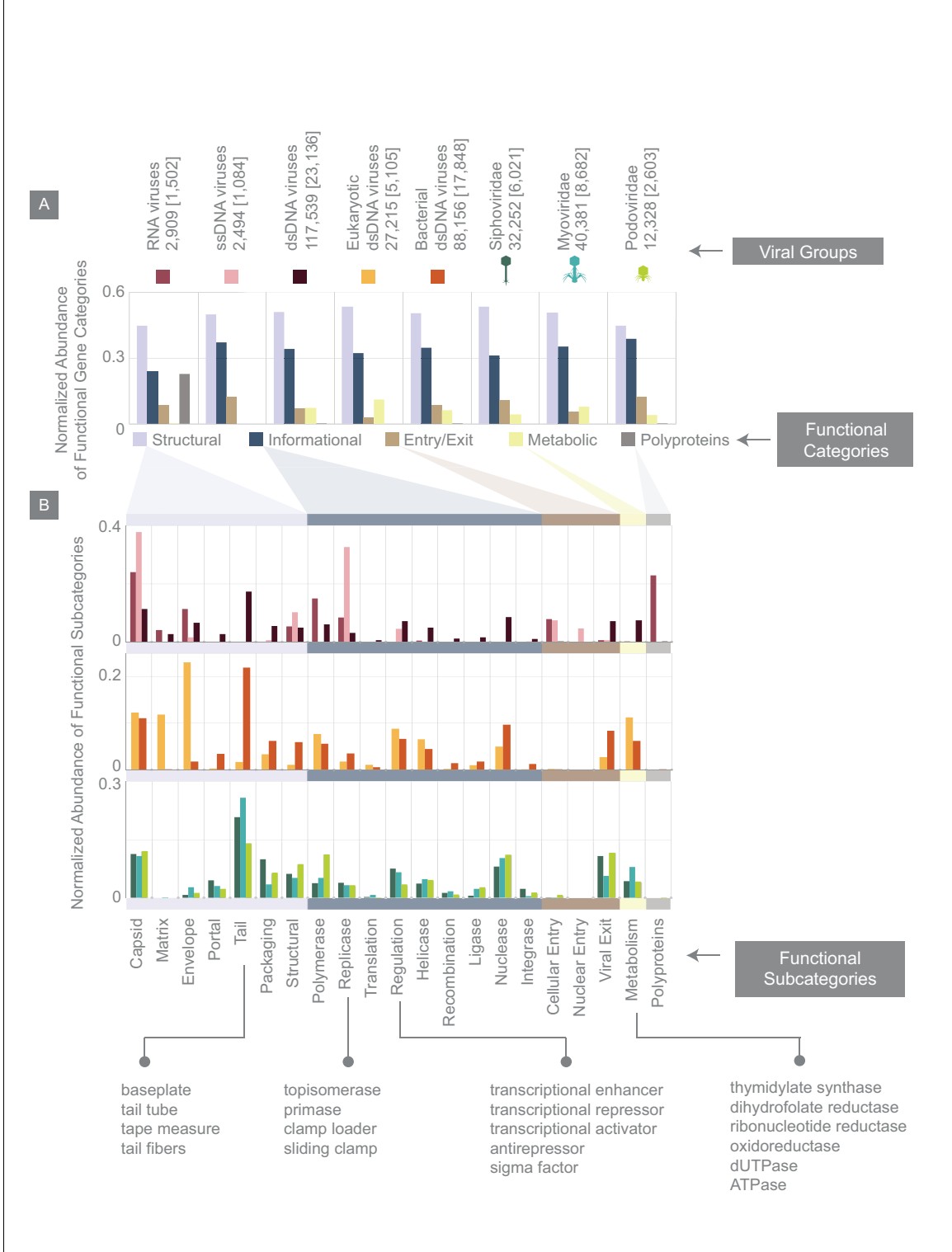

**Figure 6.** Normalized abundance of functional gene categories across different viral groups. (A) Abundances of functional gene categories across 8 viral groups normalized to the number of labeled genes in each viral group (the total number of genes in each viral group is shown above the panel, and in brackets are the number of labeled genes for each viral group). (B) Abundances of functional gene subcategories across 8 viral groups: RNA, ssDNA, and dsDNA viral groups (top plot); eukaryotic and bacterial dsDNA viral groups (middle); *Siphoviridae*, *Myoviridae*, and *Podoviridae* viral groups (bottom). A few examples of the types of genes contained as part of each functional subcategory are provided.
DOI: https://doi.org/10.7554/eLife.31955.013

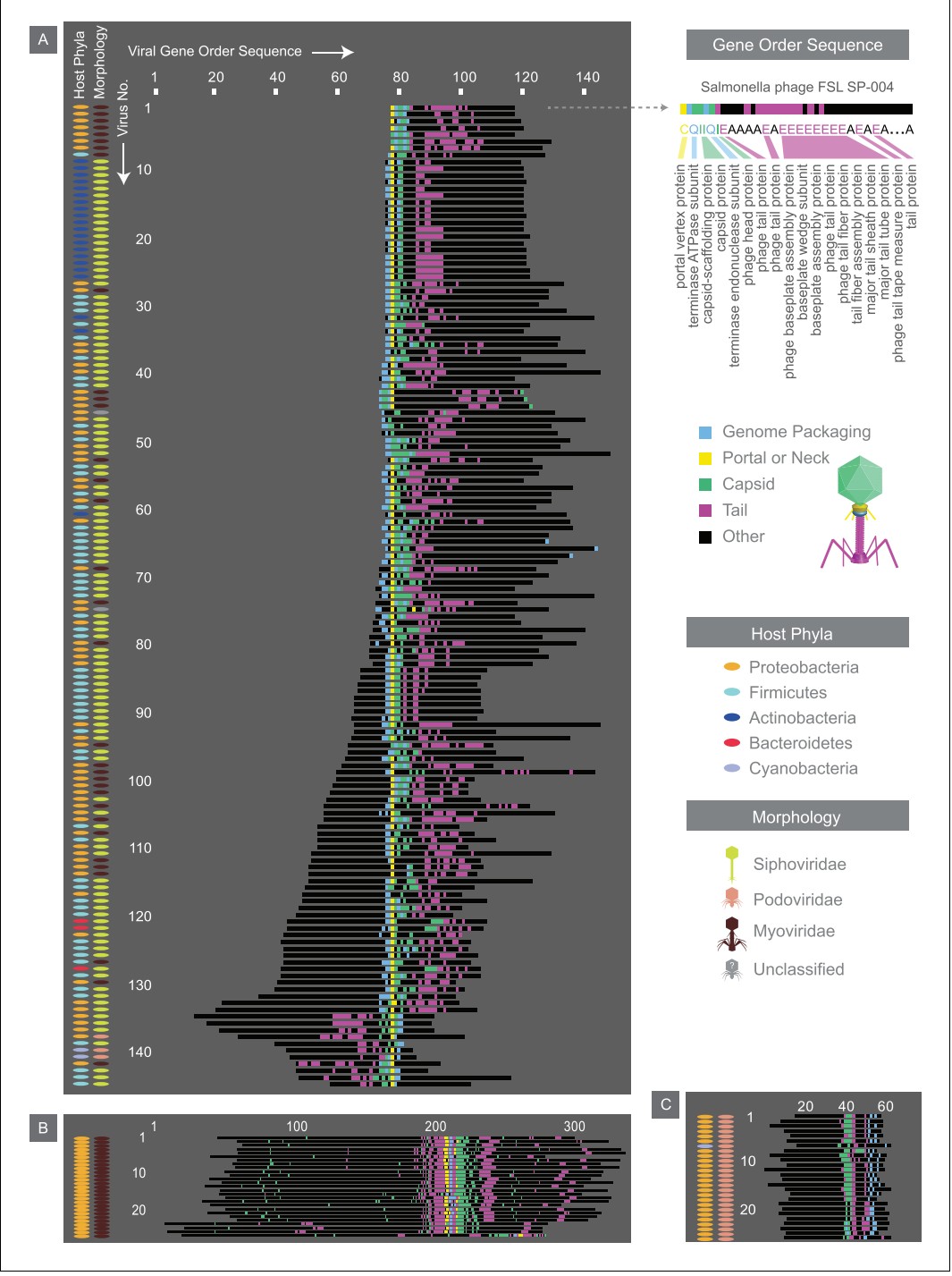

**Figure 7.** Alignment of the most common gene order patterns for dsDNA bacterial viruses. Each genome is summarized by a sequence of letters, with each letter corresponding to a gene, positioned in the order that it appears on the genome. As an example, the gene order sequence for *Salmonella phage FSL SP-004* is shown. Note the letters shown serve to only denote genes with similar functions. Structural genes are assigned colors, whereas other genes are denoted in black. Across all three panels, each row corresponds to the gene order sequence for a given virus, and thus, the length of the sequence denotes the number of genes within a given genome. The left two columns accompanying each panel provide further information on hosts and viral morphologies. Panel A, B, and C, represent gene order patterns A, B, and C, respectively. Geneious global alignment (*Steitz et al., 2011*) was used to align gene order sequences (see Materials and methods). Refer to *Figure 7 continued on next page*

*Figure 7 continued*

*Figure 7—figure supplement 1* to see the percent identity heat maps of terminases (large and small subunits) across dsDNA bacterial viruses.
DOI: https://doi.org/10.7554/eLife.31955.014
The following figure supplement is available for figure 7:

**Figure supplement 1.** Percent identity heat maps of A) 320 terminase (large subunit) amino acid sequences, and B) 191 terminase (small subunit) amino acid sequences from dsDNA bacteriophages.
DOI: https://doi.org/10.7554/eLife.31955.015

(*Mattick and Makunin, 2006*; *Morris, 2012*). With a median noncoding percentage of just 6%, RNA viral genomes have significantly lower noncoding percentage compared to DNA viruses in this database (one-sided Mann-Whitney U Test, $P<10^{-5}$). A notable exception to the RNA viral group is the ssRNA-RT with a median noncoding percentage of 16%. Interestingly, both retroviral groups had relatively high noncoding DNA percentages. This is likely due to the presence of defunct retroviral genes. For example, the *Xenopus laevis* endogenous retrovirus (NCBI taxon ID 204873) belonging to the ssRNA-RT group has a noncoding percentage of 93%. This high noncoding percentage can be explained by the fact that this virus genome contains three pseudogenes previously coding for *env*, *pol* and *gag* proteins.

### Viral functional gene categories

We categorized viral genes according to several major functional categories, including structural genes such as capsid and tail genes, metabolic genes, informational genes, which we define as those involved in replication, transcription or translation of the viral genetic code, among other categories (*Figure 6*, see Materials and methods). In addition to the fraction of viral genes that we were able to assign to these functional categories, there still remains what we will refer to as an "unlabeled" fraction that is comprised of hypothetical genes or genes with poor annotation (see Materials and methods). When reporting the relative abundance of different functional gene categories, we will normalize the number of genes belonging to each functional category by the total number of labeled genes.

RNA, dsDNA and ssDNA viruses, despite differences in the detailed categorization of their genes (*Figure 6B*) share similar general features (*Figure 6A*). For example, across all three viral groups, roughly half of all genes are structural. Similarly, dsDNA viruses of eukaryotes and bacteria in this database, in contrast to having different genomic properties and morphologies

surprisingly have very similar distribution of gene functional category and subcategory abundances. The major difference between these two viral groups, as expected from our knowledge of viral morphologies, is that a larger portion of eukaryotic dsDNA viral genes are envelope and matrix genes, whereas a greater portion of bacterial dsDNA genes are portal and tail-associated genes. By further zooming in on bacterial dsDNA viruses, it is again interesting to see that *Myoviridae*, *Siphoviridae*, and *Podoviridae* viral groups, with their different morphologies and wide range of hosts, having very similar functional gene category abundances even at the level of subcategories.

### Viral genome organization

To explore viral genome organization we developed a coarse-grained method for visualizing a large number of genomes in one snapshot. We first defined genome organization as the order in which genes appear across a genome. We then symbolized each gene by a letter, indifferent to the gene's length or its orientation on the genome. Genes with similar functions are grouped and are represented by the same letter (*Figure 7*). Therefore each viral genome, analogous to a nucleotide sequence, is compactly described by a sequence of letters that represent its gene order (*Figure 7*), which we will refer to as the gene order sequence. Because we aimed to study gene order sequences across different viral groups, we focused on genes whose functions are universally required, namely structural genes. textFile-1.txt (see our GitHub repository) provides the structural gene order sequences for all viruses (see Materials and methods for filters applied), though the script developed can be modified to visualize the placement of any number of genes or user-defined gene groups.

Furthermore, by focusing on bacterial dsDNA viruses present in the NCBI viral database, we were able to identify the most common gene order patterns across this virome (see Materials

and methods). One particular gene order pattern and its variations exist across various types of dsDNA bacterial viruses. We will refer to it as gene order pattern A (*Figure 7A*). In pattern A, gene packaging, portal and capsid-related genes are mostly tightly clustered and are followed by tail-associated genes. Interestingly, this pattern occurs at the beginning of the genome for some viruses, and for others it seems to have been shifted further down on the genome. Pattern A occurs across viruses from five different host phyla. The other two most common gene order patterns (patterns B and C)

occur across viruses with more limited host range and morphologies.

Beyond their order in the genome, we wondered to what extent are bacteriophage proteins from taxonomically similar hosts similar to each other in sequence? In an attempt to address this question, we analyzed sequences from two structural proteins in dsDNA bacteriophages, namely terminase large subunit and small subunit, which are used in the packaging of DNA inside capsids and represent some of the more clearly annotated bacteriophage proteins. Amino acid sequences were aligned using Clustal-Omega (*Arndt et al., 2016*) and the

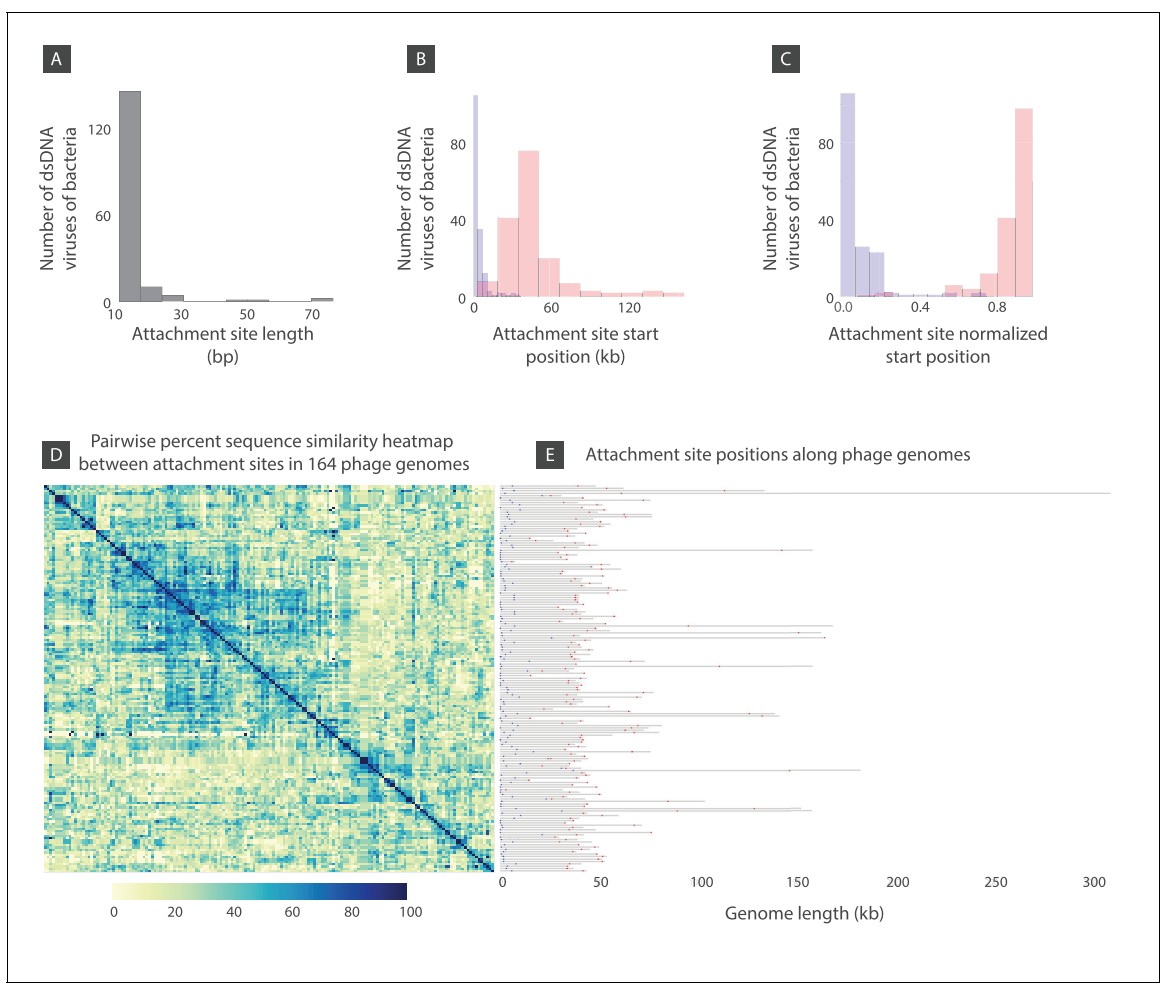

**Figure 8.** Attachment site length, position, and sequence diversity for 164 dsDNA bacterial viruses. (A) Histogram of attachment site length. (B) Histogram of attachment site start positions (left attachment: blue, right attachment: red). (C) Histogram of attachment site start positions normalized by the genome length. (D) Percent sequence similarity matrix across attachment sites. (E) Attachment site locations along viral genomes (left attachment: blue, right attachment: red). *Figure 8—source data 1* demonstrates several bacteriophages shown in panel E with similar or identical attachment site sequences.

DOI: https://doi.org/10.7554/eLife.31955.016

The following source data is available for figure 8:

**Source data 1.** Several bacteriophages from *Figure 8D* with similar or identical attachment site sequences.

DOI: https://doi.org/10.7554/eLife.31955.017

sequence similarity percentages are shown as heat maps (*Figure 7—figure supplement 1*). The host phylum information is color-code. As can be seen from this figure, bacteriophages infecting hosts from the same phylum do not necessarily have more similar terminase sequences. In the cases where there is a similarity between terminase sequences, it is primarily from bacteriophages infecting the same host species.

To provide more information on the genomic organization of dsDNA bacteriophages, we examined attachment site positions, length distributions, and sequence diversity. Attachment sites are locations of site-specific recombination that lysogenic phages use to insert their DNA into the host genome (See Materials and methods). Among the several hundred dsDNA bacteriophages that were included in this analysis, we found roughly a quarter to have putative attachment sites. We found that the median attachment site length is 13 base pairs (*Figure 8A*). The left attachment start position in the genome is located at ~2 kb (this is the median of left attachment site start positions across all genomes analyzed). The right attachment site median position is located at ~40 kb. (*Figure 8B*). *Figure 8C* demonstrates the same data but normalized by the genome length.

To examine attachment site sequence diversity, we used Clustal-Omega (*Sievers et al., 2011*) for creating a sequence alignment. *Figure 8D* is a heat map of the percent sequence similarity scores. *Figure 8E* demonstrates left (blue) and right (red) attachment sites in phage genomes. Note, the genomes are shown according to their order in *Figure 8D*. While the vast majority of attachment sites are very diverse in sequence, as shown by regions of low similarity in the heat map, there are a number of viruses that have identical putative attachment site sequences (*Figure 8—source data 1*, Materials and methods). Perhaps not surprisingly, these phages are largely those infecting different strains of the same host species. Phages infecting hosts outside of the same species seem more likely to have dissimilar attachment site sequences.

### Shedding some light on viral "hypothetical" proteins

As demonstrated in the previous sections, proteins annotated as hypothetical or putative form more than half of all proteins associated with dsDNA bacteriophages. In an attempt to learn more about these proteins, we used BLASTP to query all ~88,000 dsDNA bacteriophage proteins against the NCBI Refseq protein database (limited to bacteria) (See Materials and

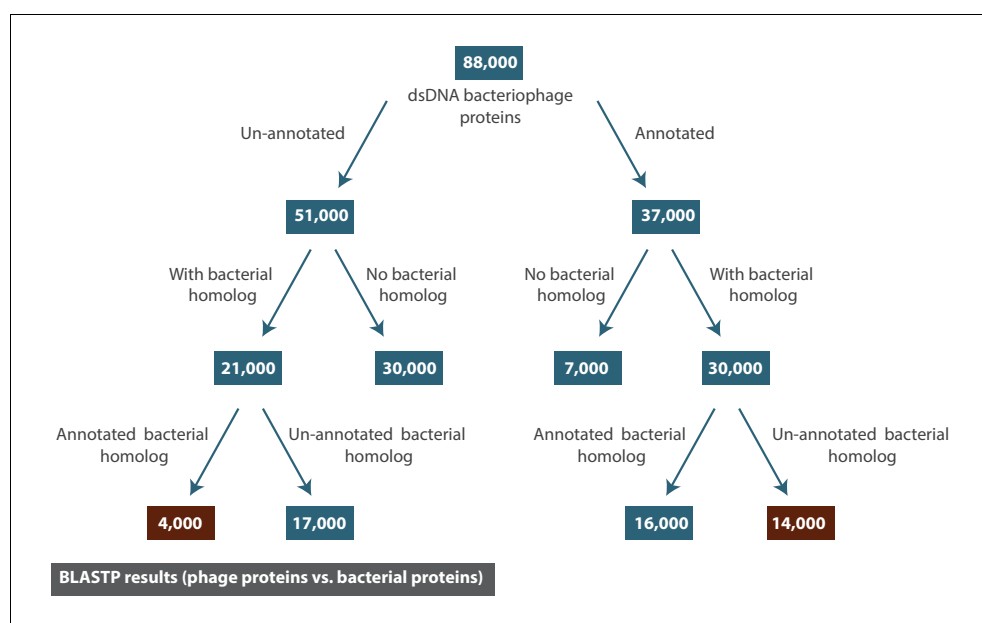

**Figure 9.** The result of BLASTP for all dsDNA bacteriophage proteins against the NCBI Refseq protein database (limited to bacterial proteins). The numbers reported correspond to the number of dsDNA bacteriophage proteins (rounded to the nearest thousand).
DOI: https://doi.org/10.7554/eLife.31955.018

methods). The purpose of this exercise was to use the annotations of bacterial homologs to viral proteins to gain better understanding of what the function of each bacteriophage hypothetical protein might be.

A homologous relationship was defined as a match with BLASTP E-value score < $10^{-10}$. The closest bacterial homolog to each bacteriophage protein (i.e. the match with the lowest E-value) was collected. Not all bacteriophage proteins had a bacterial homolog, at least not one that is currently in the NCBI database. However, a surprisingly large number did have bacterial homologs, and we have collected these proteins along with other useful information in textFile-2.txt (see our GitHub repository). This dataset is in part visualized in *Figure 9*.

Most bacterial homologs of hypothetical phage proteins were also annotated as hypothetical proteins. However, a few thousand hypothetical phage proteins could be assigned to putative annotation based on the annotation of their bacterial homologs (See Materials and methods). Interestingly, we were able to match even more bacterial hypothetical proteins to a putative annotation based on the annotations of their bacteriophage protein homologs. Although, this method can certainly be helpful in filling some of the gaps in protein annotations, it is only as good as the annotations and the convention we establish for describing proteins. Unfortunately, a considerable number of annotations are currently either too specialized or too vague to be helpful.

### The extent of overlap between viral and cellular gene pools

One of the defining features of viruses is their reliance on their host organisms. It is well known that the interactions between viruses and cells often result in the exchange of genetic information. To explore the extent to which the viral and cellular gene pools overlap, we used BLASTP to search for bacterial proteins that are homologous to dsDNA bacteriophage proteins (see Materials and methods). Overall, each of the ~900 dsDNA bacteriophage genomes we examined encoded at least one protein that was homologous to a bacterial protein.

To systematically examine the extent of homology between bacteriophage and bacterial proteins, we calculated the number of proteins per bacteriophage genome with homology to a bacterial protein, and divided this number by the total number of proteins encoded by the bacteriophage genome. In *Figure 10—figure*

*supplement 1* (left), we demonstrate the histogram of the fraction of homologous proteins per bacteriophage genome. Based on the median fraction of homologous proteins, we can conclude that 7 out of every 10 bacteriophage proteins exhibit homology to a bacterial protein. This suggests that there is a significant overlap between the two gene pools.

There are multiple mechanisms by which a bacterial protein and a bacteriophage protein could exhibit homology. The most trivial, conceptually, is when the same protein is registered as part of both a bacterial genome and a bacteriophage genome, as it would be for a prophage protein. In the case of prophages, we would expect to see a high fraction of bacteriophage proteins per genome that are homologous to bacterial proteins since their genomes should at some point in time be embedded in their hosts' genomes.

Thus, to examine the contribution from prophages, we implemented several filters to identify probable prophage genomes (see Materials and methods). Based on these filters, 173 genomes were identified. These genomes were primarily contributing to the large spike in the left histogram in *Figure 10—figure supplement 1*. To evaluate these filters, we performed a literature search for the first 20 bacteriophage genomes in the list and found that the majority were, in fact, experimentally identified as temperate phages. Because we could not find a database that contained a list of all experimentally verified prophages and their lytic relatives to compare our predictions to, we did not exclude these genomes from further analysis.

A non-trivial mechanism by which bacteriophages and bacteria can exhibit homologous proteins is via gene exchanges over evolutionary time-scales. Interestingly, the closest homolog to a bacteriophage protein is not always found in its host genome. In fact there can be large taxonomic distance (*Figure 10*) between the host and the bacterium containing the closest homolog. We depict this distance by categorizing bacteriophage proteins based on the organism in which their closest homolog was found (see inscribed circles in *Figure 10*). If they were found in the same species of bacteria as the host, then these proteins are placed in the most inner circle, whereas if they were found in the same phylum, the proteins are placed in the outer most circle.

We can see from *Figure 10* that there is a 26% chance that the closest homolog to a bacteriophage protein appears in a member of its

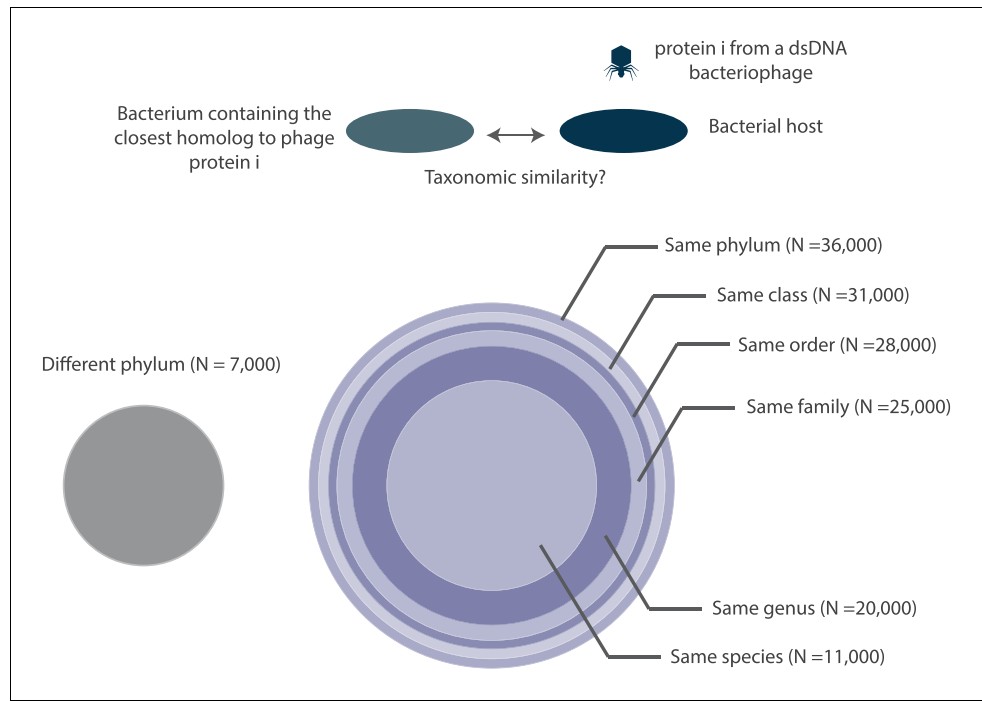

**Figure 10.** A depiction of the taxonomic distance between the bacteriophage host organism and the bacterium containing the closest homolog to a bacteriophage protein. All circles are drawn to scale with respect to the number of proteins (N) that they each represent. Note, the number of proteins denoted at each taxonomic layer includes proteins in lower taxonomic layers. For example, the 20,000 figure denoted at the genus layer already includes the 11,000 proteins shown at the species layer. N values are rounded to the nearest thousand. Histograms of the fraction of proteins with bacterial homologs per bacteriophage genome are shown in *Figure 10—figure supplement 1*.

DOI: https://doi.org/10.7554/eLife.31955.019

The following figure supplement is available for figure 10:

**Figure supplement 1.** Histogram of the fraction of proteins per bacteriophage genome with bacterial homologs (Left) and the same histogram with an additional filter to identify possible prophages and their lytic relatives (right).

DOI: https://doi.org/10.7554/eLife.31955.020

host species. This chance is raised to 84% when more broadly assuming that the homolog will appear in a bacterium that is at least in the same phylum as the host (*Figure 10*). The chance value is calculated by dividing the number of proteins in a given taxonomic layer by the total number of proteins in the analysis.

Moreover, an interesting facet of this dataset becomes apparent when we examine the quality of the match between a bacteriophage protein and its closest bacterial homolog as a function of the taxonomic distance between the bacteriophage host and the bacterium containing the homolog. We used the bit score as a measure of quality of the match. The bit score is a BLAST output and a similarity measure that is independent of database size or the query sequence length. It identifies the size of a database required for finding the same quality match by

chance. Naturally, the higher the bit score, the better is the match.

We can see that there is a significant decrease in the median bit score as we move from the "same species" layer to the "same genus" layer and finally to the "same phylum" layer (*Figure 11*). Thus, the closer (taxonomically) the host is to the bacterium containing the homolog, the better the match between the bacteriophage protein and its bacterial homolog. We think there are interesting phage-host co-evolutionary implications that can be concluded from this data analysis and data visualization method, and hope to shed further light on these hypotheses in the future.

While the majority of homologs belong to members of the same phylum as the host, there is still a 16% chance that the closest bacterial homolog to a bacteriophage protein actually

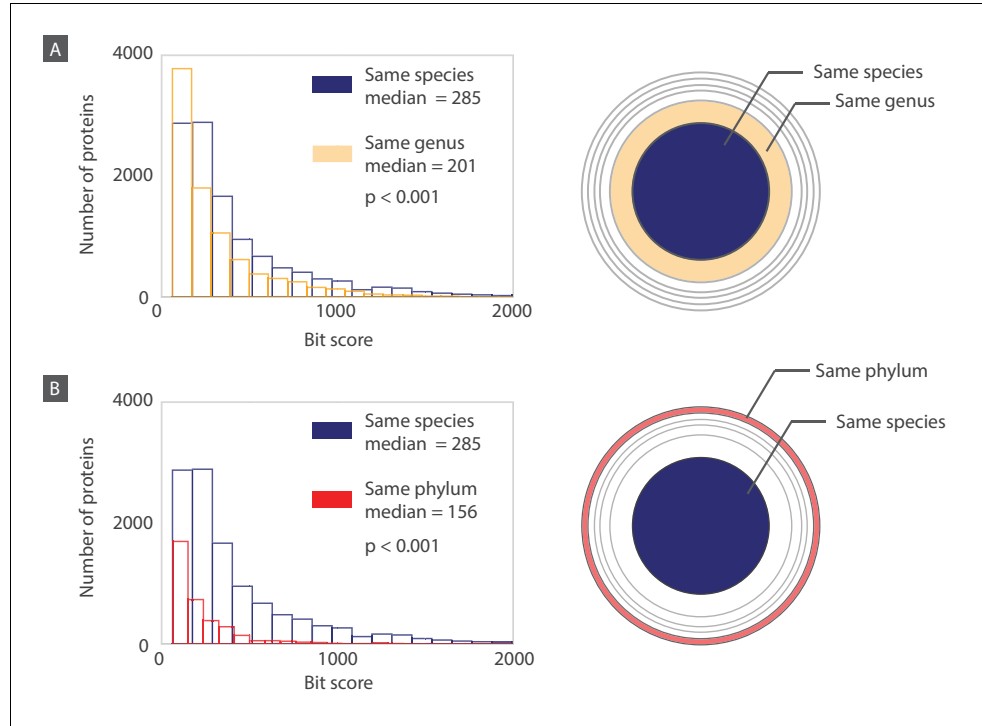

**Figure 11.** Histograms of bit scores describing the match between each bacteriophage protein and its closest bacterial homolog. Histograms are created according to the proteins belonging to three different layers corresponding to an increasing taxonomic distance between the host organism and the bacterium containing the closest homolog. (A) When the host and the homolog-containing bacterium belong to the same species, the median bit score is significantly higher (one sided Mann-Whitney U test, P<0.001) than it is for those that are only part of the same genus. (B) Similarly, when comparing proteins from the "same species" layer to the "same phylum" layer, the median bit score is significantly higher for the "same species" layer (one sided Mann-Whitney U test, P<0.001). Note that for each layer, when comparing the "same species" to the "same genus" layers, we are comparing the 11,000 proteins in the "same species" layer to the 9,000 proteins from the "same genus" layer that do not also belong to the "same species" layer. The same principle applies when we are comparing the "same species" layer to the "same phylum" layer. Distributions of bacteriophage proteins with homologs from a different phylum than their host phylum are shown in *Figure 11—figure supplement 1*.

DOI: https://doi.org/10.7554/eLife.31955.021

The following figure supplement is available for figure 11:

**Figure supplement 1.** Distributions of bacteriophage proteins with a homolog in a bacterium from a different phylum than their host phylum.

DOI: https://doi.org/10.7554/eLife.31955.022

appears in a bacterium from a different phylum than the host. To further examine these cross-phyla associations, we map the distribution of bacteriophage proteins as a function of the host phylum. Then, we zoom in on the bacterial phyla containing the homologs (*Figure 11—figure supplement 1*). By far, the most number of cross-phyla homologs are shared between bacteriophages infecting Proteobacteria and bacteria from the Firmicutes phylum. It would be interesting to explore in the future the underlying cause of the relatively large number of homologs that exist between microbial members of the Firmicutes and Proteobacteria phyla.

## Discussion

Our primary motivation for conducting a large-scale study of viral genomes was to provide the distributions of key numbers that characterize viral genomes. However, it is important to note that while the NCBI viral database represents a large collection of complete viral genomes, it still represents a small fraction of the total viral diversity in nature. In light of the striking genomic trends observed across different viral groups, future studies are needed to re-examine these trends as our databases grow in size with greater focus on several underrepresented groups such as archaeal viruses and bacterial RNA viruses. To that point, upon re-examining

the NCBI viral database in 2018, we were surprised to find that eventhough the database has almost doubled in size, the increase has disproportionately favored the already well-represented viral groups. Thus, the underrepresented groups continue to be underrepresented.

Our second motivation for conducting this study was to compare different viral classification systems. Because viral classification systems were constructed prior to the emergence of sequencing, we were interested to see how well they can describe genomic trends. Based on a comparison of classification systems across various genomic metrics, the Baltimore classification and in some cases its more minimal form (Nucleotide Type classification) seem to provide the clearest explanation for the observed trends. We suspect that this is due to the Baltimore classification's discernment of RNA, ssDNA and dsDNA genomes, which have striking physical differences.

The greater stability of dsDNA compared to RNA (*Lindahl, 1993*) and ssDNA is thought to be an important factor in the observed variations in genome lengths. The 2'-hydroxyl group in RNA makes it more susceptible to hydrolysis events and cleavage of the backbone compared to DNA. It has been shown that for bacteria and viruses, the mutation rate and the genome length are inversely correlated (*Drake, 1991*; *Sanjuán et al., 2010*), and it is therefore hypothesized that the lack of proofreading mechanisms in RNA replication and the resulting higher mutation rates compared to DNA replication (*Sanjuán et al., 2010*) imposes length limits on RNA viral genomes. In support of the suspected link between mutation rates and genome length, it has been shown that long RNA viruses (above 20 kb) contain 3'-5' exonuclease, which is a homolog of the DNA-proofreading enzymes (*Lauber et al., 2013*).

Similarly, the hydrolysis of cytosine into uracil occurs two orders of magnitude faster in ssDNA genomes than in dsDNA genomes (*Frederico et al., 1990*). This may explain the high mutation rates of ssDNA viruses, which is within the range of RNA viral mutation rates, despite using error-correcting host polymerases to replicate. In contrast to genome length in which ssDNA and RNA viruses have similar distributions, it was interesting to see that ssDNA viruses are actually more similar to dsDNA viruses in terms of their gene lengths and noncoding percentages.

While the Baltimore classification serves as a meaningful coarse-grained classification system,

it is historically animal virus centric and will benefit from being expanded to include subcategories discerning of bacterial and archaeal viruses. As shown by gene length distributions (*Figure 4*), the additional layer of categorization provided by the Host Domain classification offers new insight. For example, dsDNA and ssDNA viruses of eukaryotes have much longer gene lengths compared to their prokaryotic counterparts- an observation that may be hinting at the coevolution of host and viral genomes and proteomes since the eukaryotic genes and proteins are also shown to be significantly longer than prokaryotic ones (*Brocchieri and Karlin, 2005*; *Zhang, 2000*; *Tiessen et al., 2012*). It is well known that certain eukaryotic viral genomes, similar to their hosts' genomes, contain genes with introns (*Himmelspach et al., 1995*; *Barksdale and Baker, 1995*; *Ge and Manley, 1990*), which may explain the longer median gene length for eukaryotic viruses. In fact mRNA splicing was discovered for the first time in a study of adenovirus mRNA expression (*Flint et al., 2000*). Virus proteomes are also shown to be tuned to their hosts' proteomes by having similar codon usage and amino acid preferences (*Bahir et al., 2009*). However, future studies are needed to further ascertain the mechanisms responsible for the differences in eukaryotic and prokaryotic viral gene lengths.

The ICTV classification, which is used perhaps more than any other classification system to describe bacterial and archaeal viruses offers some supporting data (e.g. viral morphology or in some cases host information), perhaps as the final layer of classification. However, it is limited by the fact that it leaves many viruses unclassified and, more importantly, that it lacks truly systematic classification criteria. As our exploration of viruses shifts its basis from culturing of viruses to sequencing of viruses from their natural habitats, morphological data is likely to become more and more scarce. As a result, ICTV will need to adapt its classification system to operate exclusively on genomic data, a viewpoint that is broadly shared by many experts in the field (*Simmonds et al., 2017*).

In this work, we have described our attempt at providing a comprehensive and quantitative view of fully sequenced viral genomes. Similar to earlier work on biological numeracy, as exemplified by the BioNumbers database (*Milo et al., 2010*), we have identified a number of interesting trends associated with viral genomes that will be helpful in gaining a broad overview of

vastly different viral groups and their interactions with their hosts.

## Materials and methods

### Data acquisition, data curation, and statistical analysis

All genomic data was retrieved from the NCBI Genome FTP server (retrieved in August 2015) (*Brister et al., 2015*). Matching viruses to their hosts was done by parsing ASN files from the NCBI Genome FTP server while searching for the term "nat-host". All other taxonomic data, including host and viral lineages, was retrieved from NCBI's Taxonomy database using the NCBI Taxa class of the ETE Toolkit (*Huerta-Cepas et al., 2010*). Once we had the "nat-host" name of organisms in English, we retrieved their taxids using ETE Toolkit. These were in turn used to identify the host's taxonomic lineage. Viruses with complete genomes were identified by searching the assembly reports of the NCBI Genome FTP server for assemblies labeled "Complete Genome", then using the associated FTP address to download the _assembly_stats.txt files and _protein.faa files. Only viruses that could be matched to a host were included for further analysis. Additionally, various quality checks were manually performed to ensure that viruses with improper annotations were excluded from further analysis. For example, we found viruses and hosts with incomplete or incorrect taxonomic information, and excluded these viruses from further analysis. The list of excluded viruses can be found in our code (see next sub-section). Outliers are not excluded from our analysis. We attempt to dampen their effect by focusing on median values rather than the mean (*Table 1*). Given the presence of a few skewed distributions, we primarily used the Mann-Whitney U test for statistical analysis so we could avoid the assumption of normality.

#### Data availability

We have compiled all input data, output files, and scripts (Jupyter notebooks) used to write this manuscript in a GitHub repository (https://github.com/gitamahm/VirologyByTheNumbers) (*Mahmoudabadi, 2018*). viromePieChartsVF.ipynb and virusHostHistogramsVF.ipynb were used to create *Figure 2*. The scripts for *Figure 3* through *Figure 5* can be found in genomeLengthsVF.ipynb, geneLengthsVF.ipynb, and percentNoncodingVF.ipynb, respectively. The code for *Figure 6* and *Figure 7* is provided in geneOrderAndGeneAbundanceVF.ipynb. viralAttachmentSites.ipynb is used to create *Figure 8* and viralBacterialBlast.ipynb is

used to create (*Figures 9–11* and their supplementary figures).

All supplementary text files can also be found in this repository. Supplementary textFile-1.txt displays the gene order sequences for all viruses whose genomes contained at least 15% labeled genes. Letters I, C, E, and Q correspond to capsid-related, portal-related, tail-related, and genome packaging-related genes, respectively. All other genes are denoted by the letter A. Supplementary textFile-2.txt contains the list of top BLASTP matches for bacteriophage proteins that had bacterial protein homologs (the top match is considered as the match with the lowest E-value). Supplementary textFile-3.txt provides the annotations of bacterial homologs of hypothetical bacteriophage proteins. Supplementary textFile-4.txt contains the annotations of bacteriophage homologs of hypothetical bacterial proteins.

### Genome length and gene densities

Genome lengths were extracted from .ptt files and _assembly_stats.txt files for viruses. The .ptt files were parsed to find "complete genome - 1." which is followed by the length of the genome. For segmented genomes, the total length of the segments is reported as the genome length. The number of protein-coding genes, which was used in calculating gene densities, was found by parsing .faa files . For gene length histograms, we first obtained the gene lengths for each virus, and then created a histogram based on the median gene length associated with each virus. To systematically determine the number of bins needed for each histogram, we employed the Freedman-Diaconis' rule (*Freedman and Diaconis, 1981*).

### Noncoding DNA/RNA percentages

To extract the percent of the genome that is noncoding, we could not merely subtract the lengths of the genes from the length of the genome, as this would not take overlapping genes into account. Instead, we used the .ptt files to identify where each gene began and ended in the genome, then added all indices between protein-coding genes to a set. We then could subtract the size of this set from the genome length to arrive at the number of noncoding bases, which is then turned into a percentage.

### Decomposition of viral genes into functional categories

To obtain the abundance of various gene functional categories, we collected the COG product annotations (*Tatusov et al., 2000*) accompanying each gene from .ptt file(s) provided for

each virus. Based on the most frequent COG product names, we constructed a dictionary of search terms to query viral genes and measure the abundance of various functional categories (by measuring abundance, we are referring to the number of genes that belong to a given functional category). To determine the most common search terms, we derived the unique set of COG product annotations for different viromes. We used the annotations shared between viromes to exclude problematic search terms with multiple meanings. As a result we avoided search terms with multiple functional associations such as "gp41", which in the context of HIV signifies a transmembrane glycoprotein, and in the context of *Mycobacterium phage Bxb1* denotes a 3'-5' exonuclease involved in DNA replication.

While the dictionary constructed contains many key words that capture essential gene functional categories common to many viruses, it does not account for COG annotations that are non-descriptive (e.g. "phage protein" or "Z protein"). Additionally, there is typically a large number of genes that code for "hypothetical proteins". Together, these two fractions make up the unlabeled component, which we do not include for further analysis. Despite the limitations introduced by these unlabeled genes, there are still a large number of genes ($\sim 10^5$) that are included in our analysis. In constructing the relative abundances of different gene functional categories (*Figure 6*), we divide the abundance of a gene functional category by the total number of labeled genes (denoted at the top of *Figure 6*.A for each viral group).

### Gene order

In visualizing gene order we employed a similar search strategy to the one explained in the previous section. To detect potentially conserved patterns in gene order across vastly different viral genomes, we searched only for structural genes as they are essential to any virus. We used .ptt files to determine gene order since they contain the beginning and end indices of genes. The code developed uses .ptt files as input, and outputs a string of characters per viral genome, which we have referred to as the gene order sequence. Each character represents a viral gene in the order that it appears on the genome (without distinguishing between the strand of DNA on which the gene is located). All genes belonging to the same functional category, for example all tail-related genes, are represented by the same character. All unlabeled

genes (i.e. non-structural, hypothetical, or poorly annotated genes) are also represented by the same character. Each gene order sequence, analogous to a nucleotide sequence, can be aligned against other gene order sequences by existing alignment software.

Though it would be ideal to calculate a pairwise distance matrix between gene order sequences and to quantitatively define a gene order pattern based on gene order sequence similarity (akin to defining an Operational Taxonomic Unit), this effort would require the development of appropriate alignment algorithms and inference methods fit to process gene order sequences. In the meantime, we used existing alignment software as a guide and grouped gene order sequences based on generally shared features.

We used Geneious software (*Kearse et al., 2012*) to align gene order sequences using global alignment with free end gaps and identity cost matrix (with default gap open and extension penalties). Using Geneious global alignment as a guide, we further manually improved the alignment by aligning a widely shared sub-pattern, for example the portal/neck genes in pattern A or the capsid and tail characters in pattern C, without introducing any gaps. This step was necessary because any alignment algorithm will aim to maximize the alignment between unlabeled genes, unable to distinguish between these characters and the more meaningful characters corresponding to labeled structural genes. Moreover, because of the high fraction of genes that have "hypothetical protein" COG annotation, we had to impose filters to extract gene order sequences that are not entirely composed of unlabeled genes. To generate the alignments shown in *Figure 7*, we imposed that at least 15% of characters in a gene order sequence have to correspond to labeled genes, and that the gene order sequence has to be at least 40 characters long. For the gene order sequences shown in textFile-1.txt (see GitHub repository) the sequence order length limit was not imposed.

### Bacteriophage attachment sites

To explore bacteriophage attachment sites, we used the PHASTER program to obtain putative attachment sites (*Arndt et al., 2016*). Using phage genome accession numbers and the PHASTER URL API, we obtained information regarding attachment site sequence and location. We analyzed and visualized this data using our own set of scripts, which can be found in the

attachmentSites.ipynb notebook. When comparing attachment site sequences, we selected phage pairs with 100% similarity across their alignment. We also imposed that the alignment length should be at least 8 bp (which is more than half of the median attachment site length). *Figure 8—source data 1* depicts phages that met these criteria.

### BLASTing dsDNA bacteriophage proteins against bacterial proteins

Multithreaded BLASTP on this database was a computationally intensive process, requiring over 8900 core hours. Using custom scripts in R, taxID results from each protein queried were linked to complete lineages using NCBI taxdump and NCBItax2lin (available via https://github.com/zyxue/ncbitax2lin). A list of hypothetical bacteriophage proteins and their closest bacterial homologs are provided in textFile-3.txt (see our GitHub repository). Similarly the hypothetical bacterial proteins along with the annotations of their closest bacteriophage homologs are provided in textFile-4.txt. The closest homologs are determined based on the match with the lowest E-value. A match was taken into account only if it had an E-value $< 10^{-10}$.

### Identifying putative prophage genomes

We suspected that prophage proteins would have a high percent identity to their bacterial homologs, and therefore, we first filtered proteins with less than 50% identity to their bacterial homologs. We then selected only proteins with bacterial homologs if the bacterium containing the homolog was the same species as the bacteriophage host. Finally, we required that at least half of the proteins per bacteriophage genome meet the conditions described above for the bacteriophage to be identified as a potential prophage. The reason we did not impose stricter filters was so that we could also identify any lytic relatives of prophages, since their proteins would also be perceived homologous to bacterial proteins, but only because of their homology to prophage proteins.

### Acknowledgements

We thank Arup Chakraborty, Markus Covert and Richard Neher for their helpful suggestions through the review process. We would additionally like to thank Bill Gelbart, Eddy Rubin, Forest Rohwer, Eugene Shakhnovich, Matt Morgan, and members of the Phillips Lab and the Boundaries of Life Initiative for helpful discussions. We would like to especially thank Helen Foley for helping us run BLAST using Amazon cloud computing services. This study was supported by the National Science Foundation (Graduate Research Fellowship; DGE-1144469), the John Templeton Foundation (Boundaries of Life Initiative; 51250), the National Institute of Health (Maximizing Investigator's Research Award; RFA-GM-17-002), the National Institute of Health (Exceptional Unconventional Research Enabling Knowledge Acceleration; R01- GM098465), and the National Science Foundation (NSF PHY11-25915) through the 2015 Cellular Evolution course at the Kavli Institute for Theoretical Physics.

**Gita Mahmoudabadi** is in the Department of Bioengineering, California Institute of Technology, Pasadena, United States

http://orcid.org/0000-0002-8812-7246

**Rob Phillips** is in the Department of Bioengineering and the Department of Applied Physics, California Institute of Technology, Pasadena, United States

phillips@pboc.caltech.edu

http://orcid.org/0000-0003-3082-2809

**Author contributions:** Gita Mahmoudabadi, Conceptualization, Data curation, Formal analysis, Supervision, Validation, Visualization, Methodology, Writing—original draft, Project administration, Writing—review and editing; Rob Phillips, Conceptualization, Supervision, Funding acquisition, Investigation, Methodology, Project administration, Writing—review and editing

**Competing interests:** The authors declare that no competing interests exist.

### Funding

| Funder | Grant reference number | Author |
| --- | --- | --- |
| John Templeton Foundation | 51250 | Rob Phillips |
| National Institutes of Health | RFA-GM-17-002 | Rob Phillips |
| National Science Foundation | DGE-1144469 | Gita Mahmoudabadi |
| National Institutes of Health | R01- GM098465 | Rob Phillips |
| National Science Foundation | NSF PHY11-25915 | Rob Phillips |

The funders had no role in study design, data collection and interpretation, or the decision to submit the work for publication.

**Decision letter and Author response**
Decision letter https://doi.org/10.7554/eLife.31955.027
Author response https://doi.org/10.7554/eLife.31955.028

## Additional files

### Supplementary files

• Transparent reporting form
DOI: https://doi.org/10.7554/eLife.31955.023

### Major datasets

The following dataset was generated:

| Author(s) | Year | Dataset URL | Database, license, and accessibility information |
|---|---|---|---|
| Brister JR, Ako-Adjei D, Bao Y, Blinkova O | 2015 | ftp://ftp.ncbi.nlm.nih.gov/refseq/release/viral/ | Publicly available at the NCBI viral resource page (https://www.ncbi.nlm.nih.gov/genome/viruses/) |

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
