## [Decision Letter]

Thank you for submitting your article "Viral Genomes by the Numbers: A Comprehensive, Quantitative Exploration of Thousands of Viruses" for consideration by *eLife*. Your article has been reviewed by two peer reviewers, and the evaluation has been overseen by Arup Chakraborty as the Reviewing Editor and Senior Editor. The following individuals involved in review of your submission have agreed to reveal their identity: Richard A Neher (Reviewer #1); Markus Covert (Reviewer #2).

The reviewers have discussed the reviews with one another and the Reviewing Editor has drafted this decision to help you prepare a revised submission.

Summary:

Your paper provides a valuable summary of genomic features of viruses in the spirit of previous papers and collections by the senior author on "numbers in biology". Such an assessment is timely and would seem to be likely to yield insights of potentially broad interest. You characterize genome length, gene length, gene density, portions of the genome not coding for proteins, and gene order. Among other interesting details, the paper makes a strong argument at the beginning about the limitations of metagenomic analysis and about the utility of studying whole genomes. We appreciated the thorough quantification of several physical metrics, and one figure in particular which showed the organization of viral genomes in relation to each other – which is likely to have great potential for interesting discoveries. You also investigate the degree to which different classification schemes explain statistical trends in quantitative characters.

Based on our consultation and the reviews summarized below, we propose that you consider one of two possible paths forward.

1) Address the specific points below and the comments provided for the second path forward, if the paper is to be considered as a resource.

- Provide all relevant files to make the scripts usable.

- Update the analysis/data bank with genomes that have become available since. The archea set is tiny, and the phage set small. Many additional genomes have likely become available since. Update could be selectively with genomes from underrepresented categories (e.g. from 10.7554/*eLife*.08490).

- Provide a web interface, or something similar, to access the data/analysis.

2) Alternatively, if you cannot address the points above easily, we would be happy to consider publication of your valuable contribution as a "feature article", provided you are able to address most of the points below.

- While the Baltimore classification system stratifies genomic features well, it is quite animal virus centric. Are there better systems for phages/archael viruses?

- Given the paper's mention of viral dark matter, the fraction of putative viral genes in the database that are not annotated seems important. An attempt to classify them (e.g. BLASTing them against the host's genome, see also the next comment) would also be recommended.

- While this may be prohibitive, one could explore the correlation between viral genome size with virus size if the physical size for the viruses is present in the database.

- Comparative analysis of the location and sequence of any known attachment sites.

Exploration of the host context could be very helpful to the paper. For example, the authors could identify, categorize and locate host genes in the viral genome. A similar analysis was recently done in bacteria (https://www.nature.com/articles/s41467-017-00808-w) and led to very surprising results. An additional suggestion might be to quantify the sequence similarity of viral structural elements within and between host categories (i.e. # of nucleotides and aa different after alignment, etc.).

- The language of the paper should be modified to make it clear that the statistical quantification reflects properties of the database, not what is present in natural environments. While the database contains a significant ~2500 viruses, this is in fact a small number in comparison to species present in natural environments. A good example of how this can skew perceptions of abundance is found in Young et al. (http://www.pnas.org/content/113/37/10400.full), where they showed 91% of sequencing reads arose from just 7 viral contigs. We recommend that you include some mention of this in the Discussion.

---

## [Author Response]

Based on our consultation and the reviews summarized below, we propose that you consider one of two possible paths forward.

Thank you very much for the excellent feedback, and for providing us extensions to be able to complete this rebuttal. We enjoyed these suggestions so much that we delved deeply into each topic. We have now added 4 new subsections, 7 new figures, 1 table, several new python scripts, and assembled several large datasets (one of which, for example, took 8900 core hours to assemble). We would like to be considered for the second path to publication. For the sake of completion, we also responded to the additional suggestions under the first path to the extent that we could given our time constraints.

1) Address the specific points below and the comments provided for the second path forward, if the paper is to be considered as a resource.- Provide all relevant files to make the scripts usable.

All raw data files, scripts, and relevant output files are provided in the Virology by the Numbers GitHub repository which can be accessed from:

https://github.com/gitamahm/VirologyByTheNumbers

- Update the analysis/data bank with genomes that have become available since. The archea set is tiny, and the phage set small. Many additional genomes have likely become available since. Update could be selectively with genomes from underrepresented categories (e.g. from 10.7554/eLife.08490).

Due to time constraints, we were unfortunately unable to re-run all the analysis on the now larger NCBI viral database of complete genomes. We hoped to selectively update under-represented categories, but we have come to the conclusion that the increase in the number of genomes has disproportionately favored the already well-represented categories. For example, there are still no RNA or ssDNA archaeal viruses reported to the database. The number of dsDNA archaeal viruses has increased to 76 viruses, which is not much of an improvement to our set of 46. Other under-represented groups are RNA and ssDNA viruses of bacteria, which we again saw not much of an improvement on (for example 88 ssDNA viruses of bacteria compared to our set of 51, and the 88 figure could be lowered after we perform quality control checks on various annotations). We would like to perform an update in the future, preferably when there are an order magnitude more viruses with complete genomes available.

We have added a short statement of this observation in the Discussion section.

- Provide a web interface, or something similar, to access the data/analysis.

All raw datasets, scripts, and output files are provided in the GitHub repository associated with this manuscript: https://github.com/gitamahm/VirologyByTheNumbers

2) Alternatively, if you cannot address the points above easily, we would be happy to consider publication of your valuable contribution as a "feature article", provided you are able to address most of the points below.- While the Baltimore classification system stratifies genomic features well, it is quite animal virus centric. Are there better systems for phages/archael viruses?

In the Discussion section, we now further describe the Baltimore classification as being animal virus-centric and mention that the primary classification system used for bacterial and archaeal viruses is the morphology-driven ICTV classification. We further describe in this section the need for genomic-based classification for phages and other types of viruses.

- Given the paper's mention of viral dark matter, the fraction of putative viral genes in the database that are not annotated seems important. An attempt to classify them (e.g. BLASTing them against the host's genome, see also the next comment) would also be recommended.

Thank you for this interesting suggestion. We have described our findings in a new subsection “Shedding some light on viral “hypothetical” proteins” as well as a new figure (Figure 9).

We focused our attention on dsDNA viruses of bacteria (or dsDNA bacteriophages), as they have the largest fraction of hypothetical proteins than any other virus group. All dsDNA bacteriophage proteins were queried against the NCBI Refseq protein database (limited to bacteria) in an attempt to classify/categorize these proteins and to find out more information on the fraction of proteins that were labeled as hypothetical. Multithreaded BLASTP on this database was a computationally intensive process, requiring over 8900 core hours. The closet bacterial homolog to each bacteriophage protein (i.e. the match with the lowest E-value, requiring a threshold E-value < 10^-10^ for the match between any two proteins to be considered significant) was collected. Not all bacteriophage proteins had a bacterial homolog, at least not one that is currently in the NCBI database. However, a surprisingly large number (~51,000 proteins) did have bacterial homologs, and we have collected these proteins along with other useful information in bactBlastOutputAnalyzedTopHits.txt.

We found ~21,000 dsDNA bacteriophage hypothetical proteins that were homologous to a bacterial protein, representing about a quarter of all phage proteins. Most bacterial homologs of hypothetical phage proteins were also annotated as hypothetical proteins. However, ~4000 hypothetical dsDNA bacteriophage proteins could be assigned to putative annotation based on the annotation of their bacterial homologs. These proteins and their homolog’s notations are provided in hypoPhageProtCharacterized.txt (see the GitHub repository).

Interestingly, through this exercise we were also able to match ~14,000 bacterial hypothetical proteins to a putative annotation based on the annotations of their bacteriophage protein homologs. These bacterial proteins along with the annotations of their homologs are provided in hypoBactProtCharacterized.txt. Although, this method can certainly be helpful in filling some of the gaps in protein annotations for both bacterial and phage genomes, it is only as good as the annotations and the convention we establish for describing proteins. For example, a considerable number of annotations are currently either too specialized to a particular field (e.g. “gp40”) or too vague (e.g. “phage protein”) to be helpful. The scripts written to answer this question are provided under the viralBacterialBlast.ipynb notebook (see the GitHub repository).

- While this may be prohibitive, one could explore the correlation between viral genome size with virus size if the physical size for the viruses is present in the database.

This has been an important topic to us and we had actually attempted to investigate the relationship between genome size and capsid size in the beginning of this study, but we found that there was already a recent paper (1) that provides this data using VIPER – the largest database on viral capsid sizes (2). We now have included this reference anddescribed its main findings under the “Viral genome lengths, gene lengths and gene densities” subsection. To summarize, they demonstrate that there is no significant relationship between capsid volume and genome length. By examining a diverse collection of viruses from the NCBI database, representing members of different Baltimore-classified viral groups, they conclude that most viruses use the capsid volume sub-optimally.

- Comparative analysis of the location and sequence of any known attachment sites.

We have now included several paragraphs in the “Viral genome organization” section along with a new figure (Figure 8). We wrote attachmentSites.ipynb to analyze the data (see GitHub repository). We have also added a new table (Figure 8—source data 1).

To answer this question we first searched for known attachment sites, but realized that there are not many that have been experimentally determined so far. We then aimed to include putative attachment sites in our analysis. To arrive at putative attachment site locations and sequences in bacteriophage genomes, we used the PHASTER program (3). Among the 600+ dsDNA viruses of bacteria that were included in this analysis, we found roughly a quarter to have putative attachment sites. We found that the median attachment site length is 13 base pairs (Figure 8A). The left attachment start position in the genome is located at ~2 kb (this is the median value of left attachment site start positions across all genomes analyzed). The right attachment start site median position occurs at ~40 kb (Figure 8B). Figure 8C demonstrates the same data but normalized by the phage genome length.

To examine attachment site sequence diversity, we used Clustal-Omega (4) for creating a sequence alignment. Figure 8D is a heatmap of the percent sequence similarity scores. Figure 8E demonstrates left (blue) and right (red) attachment sites in phage genomes. Note, the genomes are shown according to their order in Figure 8D. When comparing attachment site sequences, we selected phage pairs with 100% similarity across their alignment. We also imposed that the alignment length should be at least 8 bp (which is more than half of the median attachment site length). Figure 8—source data 1 depicts phages that met these criteria. While the vast majority of attachment sites are very diverse, as shown by regions of low similarity in the heatmap, there are a number of viruses that have identical putative attachment site sequences.

Perhaps not surprisingly, these phages are largely those infecting different strains of the same host species. Phages infecting hosts outside of the same species seem more likely to have dissimilar attachment site sequences.

Exploration of the host context could be very helpful to the paper. For example, the authors could identify, categorize and locate host genes in the viral genome. A similar analysis was recently done in bacteria (https://www.nature.com/articles/s41467-017-00808-w) and led to very surprising results.

We describe our findings in a new subsection “the extent of overlap between viral and cellular gene pools”, along with new figures (Figure 10, Figure 10—figure supplement 1, Figure 11 and Figure 11—figure supplement 1).

To identify, locate and categorize potential host genes in viral genomes, we used BLASTP to search for bacterial proteins that are homologous to bacteriophage proteins (homology was strictly defined as any match with an E-value < 10^-10^). Each of the ~900 dsDNA bacteriophage genomes we examined encoded at least one protein that was homologous to a bacterial protein. To systematically examine the extent of homology between bacteriophage and bacterial proteins, we calculated the number of proteins per bacteriophage genome with homology to a bacterial protein, and divided this number by the total number of proteins encoded by the bacteriophage genome. In Figure 10—figure supplement 1 (left), we demonstrate the histogram of the fraction of homologous proteins per bacteriophage genome for ~900 genomes. Based on the median fraction of homologous proteins per bacteriophage genome, about 7 out of every 10 proteins exhibits homology to a bacterial protein. This suggests that there is a significant overlap between the two gene pools.

There are multiple mechanisms by which a bacterial protein and a bacteriophage protein could exhibit homology. The most trivial, conceptually, is when the same protein is registered as part of both a bacterial genome and a bacteriophage genome, as it would be for a prophage protein. In the case of prophages, we would expect to see a high fraction of bacteriophage proteins per genome that are homologous to bacterial proteins since their genomes should at some point in time be embedded in their hosts’ genomes. Thus, to examine the contribution from prophages, we implemented several filters to identify probable prophage genomes. We suspected that prophage proteins would have a high percent identity to their bacterial homologs, and therefore, we first filtered proteins with less than 50% identity to their bacterial homologs. We then selected only proteins with bacterial homologs if the bacterium containing the homolog was the same species as the bacteriophage host. Finally, we required that at least half of the proteins per bacteriophage genome meet the conditions described above for the bacteriophage to be identified as a potential prophage. The reason we did not impose stricter filters was so that we could also exclude any lytic relatives of prophages, since their proteins would also be perceived homologous to bacterial proteins, but only because of their homology to prophage proteins.

Based on these filters, 173 genomes were identified. These genomes were primarily contributing to the large spike in the left histogram (Figure 10—figure supplement 1). As a sanity check, we performed a literature search for the first 20 bacteriophage genomes in the list and found that the majority were experimentally identified as temperate phages. However, because we could not find a database that contained a list of all experimentally verified prophages to compare our predictions to, we did not exclude these genomes from any further analysis.

A non-trivial mechanism by which bacteriophages and bacteria can exhibit homologous proteins is via gene exchanges over evolutionary time-scales. According to our analysis, there is a 26% chance that the closest homolog to a bacteriophage protein appears in its host or in members of the host species. This chance is raised to 84% when more broadly assuming that the homolog will appear in an organism within the same phylum as the host organism (Figure 10).

An interesting facet of this dataset becomes apparent when we examine the bit score as a measure of “the goodness of match” between each bacteriophage protein and its bacterial homolog. When we create histograms of bit scores for proteins at different layers of the inscribed circles (Figure 10), we can see that there is a significant decrease in the median bit score as we move from the “same species” layer to the “same genus” layer and finally to the “same phylum” layer (Figure 11). In other words, the closer (taxonomically) the host is to the bacterium containing the homolog, the better the match between the bacteriophage protein and its bacterial homolog can expected to be. We think there are interesting phage-host co-evolutionary implications that can be concluded from this data analysis and data visualization method, and hope that others could shed further light on these hypotheses in the future.

While the majority of homologs belong to the “same phylum” category, there is still a 16% chance that the closest homolog actually appears in an organism from a different phylum than the host. To further examine these cross-phyla homologs, we map the distribution of bacteriophage proteins as a function of the host phylum. Then, we zoom in on the phyla of bacteria containing the homologs (Figure11—figure supplement 1). By far, the most number of cross-phyla homologs are shared between bacteriophages infecting Proteobacteria and members of the Firmicutes phyla.

An additional suggestion might be to quantify the sequence similarity of viral structural elements within and between host categories (i.e. # of nucleotides and aa different after alignment, etc.).

We describe our findings in “Viral genome organization” subsection and in a new Figure (Figure 7—figure supplement 1).

In our attempt to address this question, we analyzed sequences from two structural proteins in the dsDNA bacteriophages, namely terminase large subunit and small subunit, which are used in the packaging of DNA inside capsids. Amino acid sequences were aligned using Clustal-Omega and the sequence similarity percentages are shown as heatmaps (Figure 7—figure supplement 1). The host phylum information is color-coded and shown as well. As can be seen from this figure, bacteriophages infecting hosts from the same phylum do not necessarily have more similar terminase sequences. In the cases where there is a similarity between terminase sequences, it is primarily from bacteriophages infecting the same host species.

- The language of the paper should be modified to make it clear that the statistical quantification reflects properties of the database, not what is present in natural environments. While the database contains a significant ~2500 viruses, this is in fact a small number in comparison to species present in natural environments. A good example of how this can skew perceptions of abundance is found in Young et al. (http://www.pnas.org/content/113/37/10400.full), where they showed 91% of sequencing reads arose from just 7 viral contigs. We recommend that you include some mention of this in the Discussion.

This is an important point, and we completely agree. We have now devoted the first paragraph of the Discussion to further emphasize this point. We have also revised our language in 8 different parts of the manuscript to make it clear that the viral genomic trends that we see are representative of viruses in the NCBI viral database and not all viruses at large.

Rebuttal References:

1) Brandes N and Linial M (2016) Gene overlapping and size constraints in the viral world. Biology direct 11(1):1.

2) Shepherd CM, et al. (2006) VIPERdb: a relational database for structural virology. Nucleic acids research 34(suppl 1):D386-D389.

3) Arndt D, et al. (2016) PHASTER: a better, faster version of the PHAST phage search tool. Nucleic acids research 44(W1):W16-W21.

4) Sievers F, et al. (2011) Fast, scalable generation of high‐quality protein multiple sequence alignments using Clustal Omega. Molecular systems biology 7(1):539.